# A metagenomic catalog of the early-life human gut virome

Shuqin Zeng[1,2], Alexandre Almeida [3], Shiping Li[1,2], Junjie Ying[1,2], Hua Wang[1,2], Yi Qu[1,2], R. Paul Ross [4], Catherine Stanton[4,5], Zhemin Zhou [6], Xiaoyu Niu [2,7] ✉, Dezhi Mu [1,2] ✉ & Shaopu Wang [1,2] ✉

Early-life human gut microbiome is a pivotal driver of gut homeostasis and infant health. However, the viral component (known as "virome") remains mostly unexplored. Here, we establish the Early-Life Gut Virome (ELGV), a catalog of 160,478 non-redundant DNA and RNA viral sequences from 8130 gut virus-like particles (VLPs) enriched or bulk metagenomes in the first three years of life. By clustering, 82,141 viral species are identified, 68.3% of which are absent in existing databases built mainly from adults, and 64 and 8 viral species based on VLPs-enriched and bulk metagenomes, respectively, exhibit potentials as biomarkers to distinguish infants from adults. With the largest longitudinal population of infants profiled by either VLPs-enriched or bulk metagenomic sequencing, we track the inherent instability and temporal development of the early-life human gut virome, and identify differential viruses associated with multiple clinical factors. The mother-infant shared virome and interactions between gut virome and bacteriome early in life are further expanded. Together, the ELGV catalog provides the most comprehensive and complete metagenomic blueprint of the early-life human gut virome, facilitating the discovery of pediatric disease-virome associations in future.

A healthy gut microbiome is important for optimum health throughout life. In particular, the early-life human gut microbiome plays a vital role in the maturation of the gut microbiome, the immune system, and overall health early and later in life[1–3]. Disturbances of the early-life human gut microbiome have been implicated in different pediatric diseases, such as necrotizing enterocolitis, inflammatory bowel diseases, malnutrition, and obesity[4,5]. Until now, the bacteria colonizing the human gastrointestinal tract (termed the "bacteriome") have been intensively studied from infancy to adulthood and to old age, and a broad picture of the composition and metabolic functions of the bacteriome as well as its disease-associated biomarkers have been extensively inferred[6–8].

Within the gut microbiome ecosystem, apart from the bacterial component, an immense number of viruses (termed the "virome") are present and profoundly interact with the other microbial communities, including bacteria, archaea, and eukaryotes[9–12]. Bacteriophages (or phages) constitute the majority of the gut virome and infect the bacteria in a lytic and/or temperate-specific manner, thus being involved in an interactive network between the bacteriome and human health[4,13,14]. While the gut viruses colonize the human host from birth

[1]Department of Pediatrics, West China Second University Hospital, Sichuan University, Chengdu, China. [2]Key Laboratory of Birth Defects and Related Diseases of Women and Children (Sichuan University), Ministry of Education, West China Second University Hospital, Sichuan University, Chengdu, China. [3]Department of Veterinary Medicine, University of Cambridge, Cambridge, UK. [4]APC Microbiome Ireland, University College Cork, Cork, Ireland. [5]Teagasc Food Research Centre, Moorepark, Fermoy, Co. Cork, Ireland. [6]Pasteurien College, Medical College of Soochow University, Soochow University, Suzhou, China. [7]Department of Obstetrics and Gynecology, West China Second University Hospital, Sichuan University, Chengdu, China. ✉e-mail: niuxy@scu.edu.cn; mudzh@scu.edu.cn; shaopu.wang@scu.edu.cn

simultaneously with the bacteriome[15], our knowledge of the dynamics of the virome and their interactions with the bacteriome early in life are largely lagging behind. Thus, further studies are needed to fully decipher the intricate and mutualistic relationship within the human gut microbiome and their impacts on health and disease.

Recent studies have improved our understanding of the gut virome composition by generating comprehensive viral genome databases, such as the Gut Virome Database (GVD)[16], the Gut Phage Database (GPD)[17], the Metagenomic Gut Virus (MGV) catalog[18], and the Cenote Human Virome Database (CHVD)[19] using either large-scale bulk metagenomes or virus-like particles (VLPs) enriched metagenome sequencing. These databases not only revealed a number of unknown viral genomes that expanded the diversity of the human gut virome, but can now be used as informative annotation resources for future alignment-based studies. However, all these databases were mainly reconstructed with adult fecal metagenomes, resulting in a lack of representation of early-life viral sequences and limiting their application in the early-life human gut virome research.

Noteworthily, since the first report on the gut virome of a newborn published over a decade ago[20], specific properties of the early-life human gut virome in comparison to that of adults have been noted due to their rapid evolution, including the total VLPs, the diversity, the dominant taxa, and the dynamic succession of the gut virome[15,21]. Nevertheless, the origin of early-life human gut microbiome remains a matter of debate. The microbes shared by mothers and infants has been well described at different taxonomic resolutions and microbial genes, however most studies thus far have solely focused on the bacteria[22–25]. It was shown that the maternal gut bacteriome exhibited higher prevalence and abundance in the early-life human gut bacteriome, exhibiting stable colonization in the gastrointestinal tract of their offspring compared to strains from other sources[26]. Similarly, a few studies have attempted to identify the shared viruses by mothers and their offspring, revealing a number of pioneering early-life human gut phages potentially deriving from the mothers[27,28]. However, given the limited sample size of mother-infant dyads, to what extent the mother-infant shared viruses contribute to the overall composition of the early-life human gut virome and the shaping factors remain largely unanswered.

Here, we established the Early-Life Gut Virome (ELGV) catalog of humans, a genomic database containing 160,478 non-redundant viral sequences curated from 8130 fecal VLPs-enriched or bulk metagenomic samples collected from human subjects over the first three years of life. Of note, the ELGV represents over 82,000 candidate viral species, providing a unique perspective into the composition and longitudinal succession of the early-life human gut virome and the contribution of shaping factors. We further extracted viral sequences from fecal metagenomic samples from mothers of included infants and additionally unrelated adults to examine the shared and unique viruses from mother-infant pairs or the early life of humans. We thus expect that the ELGV catalog will pave the way as a resource for better viral discovery and further research towards uncovering the hidden associations of the virome with health and disease in early life.

## Results

### Reconstruction and characterization of the ELGV catalog covering DNA and RNA viruses

To comprehensively characterize the human gut virome early in life, we analyzed 8130 public fecal metagenomes from infants under three years old, including both 1865 VLPs-enriched and 6265 bulk metagenomes spanning 35 studies with a collection of multiple clinical factors that have been well-documented to influence the composition of early-life human gut bacteriome, including delivery mode, gestational age at birth, and feeding pattern at sampling (Fig. 1a; Supplementary Data 1).

Given the challenge of identifying viruses in fecal metagenomic samples due to their low abundance, we employed three independent viral identifiers with different algorithms, including VirFinder[29], VIBRANT[30], and VirSorter2[31] (see "Methods"). The obtained viral sequences ($n = 3,375,049$) were then combined per sample and quality checked by CheckV[32], resulting in 625,512 viral sequences longer than 3 kbp. The filtered viral sequences were dereplicated across fecal samples and studies into 160,498 DNA and RNA viral sequences.

To estimate how many viral operational taxonomic units (vOTUs) our catalog represented, the ELGV catalog was clustered at 95% average nucleotide identity (ANI) over 85% alignment fraction (AF) of the shorter sequence[33]. This resulted in 82,152 vOTUs corresponding approximately to a species-level clustering, which were then taxonomically annotated as described below in detail. Notably, we found that 11 vOTUs were assigned to the viral families of *Phycodnaviridae* ($n = 5$ with eight viral sequences), *Mimiviridae* ($n = 4$ with eight viral sequences), and *Marseilleviridae* ($n = 2$ with four viral sequences), which likely represent contaminants or misclassifications, as previously suggested[16,34]. Therefore, we manually removed these vOTUs and their viral sequences, resulting in 82,141 vOTUs containing 160,478 viral sequences, henceforth referred to as the ELGV catalog (Fig. 1b; Supplementary Data 2 for viral sequences, Supplementary Data 3 for vOTU representatives). Among them, 27.0% vOTU representatives were obtained from VLPs-enriched metagenomes, and 13.1% vOTU representatives were categorized as high quality based on the MIUViG quality tiers[33] (Fig. 1c). The vOTU representatives had a median length of 8281 bp (interquartile range, IQR = 4,579–20,650 bp), and 36,059 vOTU representatives had a length >10 kbp. Among these, 10,787 vOTU representatives (containing $n = 61,192$ viral sequences) were categorized to be complete or high-quality (>90% completeness), 10,508 vOTU representatives ($n = 23,804$ viral sequences) were estimated to be medium-quality (50–90% completeness), 60,787 vOTU representatives ($n = 75,399$ viral sequences) were low-quality genomes (<50% completeness), and 59 vOTU representatives ($n = 83$ viral sequences) as "not-determined" (Fig. 1b). The proportion of the proviruses was 7.80%, and the majority of vOTU representatives were with temperate lifestyle accounting for 68.4% (Fig. 1c).

To explore the taxonomic annotation of vOTUs, we first compared the vOTU representatives to the viral proteins from the UniProt Knowledgebase (UniProtKB) and then assigned each vOTU representative at the family level that is most prominently used[15,18] with the voting approach by Demovir (see "Methods"). A total of 46 viral families were annotated covering 52,224 vOTUs containing 119,126 viral sequences, and the other vOTUs (36.4% of total) without family annotations were either without a hit to UniProtKB ($n = 28,694$, termed as the "unmatched") or failed to assign a family name due to the voting approach ($n = 1223$, termed as the "unassigned"), highlighting the incompleteness and knowledge gap of the current taxonomy of early-life human gut virome. Among the taxonomically assigned vOTUs in the ELGV catalog, 99.6% ($n = 52,030$ vOTUs) were bacteriophages, and the other vOTUs (0.4%, $n = 194$ vOTUs) represented eukaryotic viruses (Supplementary Data 3). Most of the vOTUs were annotated to *Siphoviridae* family ($n = 40,491$ with 92,661 viral sequences), followed by *Myoviridae* ($n = 5690$ with 13,720 viral sequences), *Peduoviridae* ($n = 2024$ with 4777 viral sequences), *Podoviridae* ($n = 1409$ with 3303 viral sequences), and *Microviridae* ($n = 1046$ with 1636 viral sequences) (Fig. 1d). Three RNA viral families, including *Caliciviridae* ($n = 14$ with 16 viral sequences), *Picornaviridae* ($n = 10$ with ten viral sequences), and *Astroviridae* ($n = 3$ with three viral sequences), were detected. In parallel, we also adopted the approach from ref. 18 to cluster the species-level vOTU representatives into higher ranking clades (i.e., genus and family levels) based on the pairwise average amino acid identity (AAI) and gene sharing, which resulted in 11,413 genus-level vOTUs and 1238 family-level vOTUs. Rarefaction analysis of the total number of viral

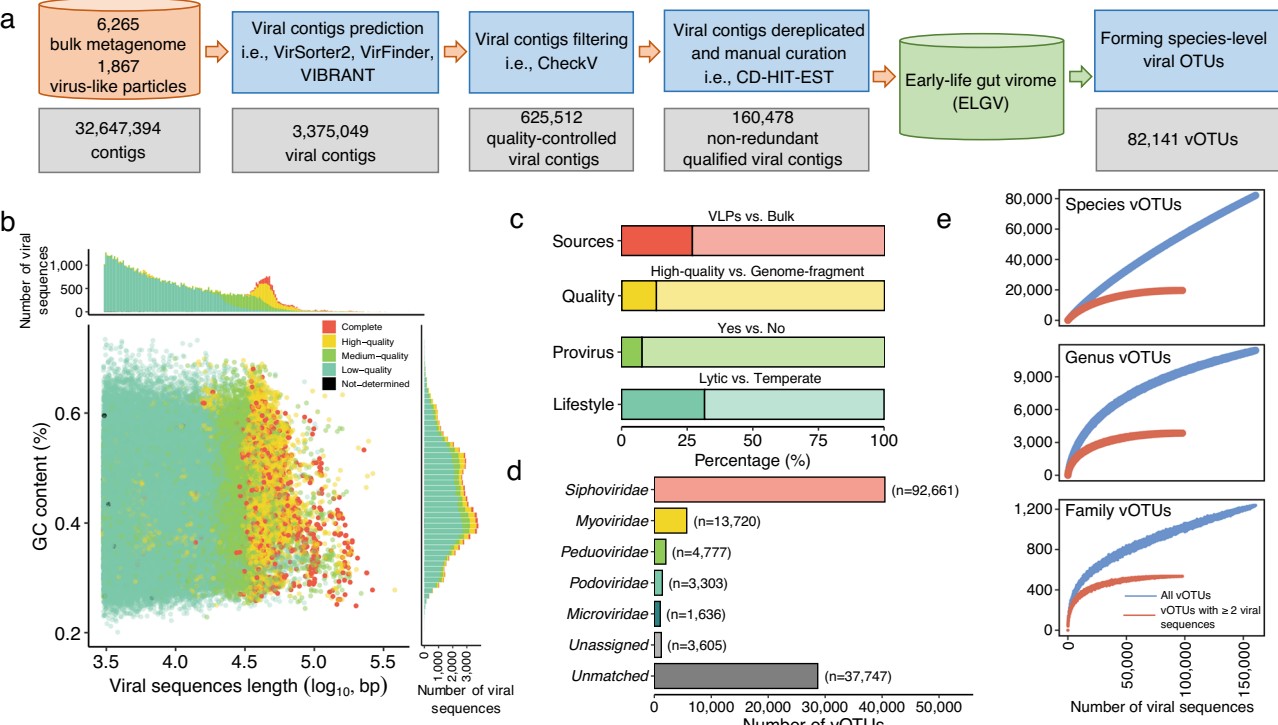

**Fig. 1 | Reconstruction and characterization of the early-life human gut virome catalog. a** The pipeline to establish the early-life human gut virome catalog with 160,478 viral sequences, representing 82,141 viral OTUs. **b** The distribution of 82,141 vOTU representatives plotted by their length and GC content. Bar plots on the top and right side show the distribution of the length and GC content, respectively. The points are colored according to the quality of viral sequences assessed by CheckV. **c** Characterization of 82,141 vOTU representatives. **d** Taxonomic annotation of 82,141 viral OTUs. The number in parenthesis indicates the number of viral sequences. **e** Accumulation curves for the early-life human gut virome catalog at the species-, genus- and family levels. The blue curves are plotted with inclusion of all vOTUs (n = 82,141), while the red curves are plotted only with vOTUs (n = 19,604) consisting of at least two conspecific viral sequences.

sequences against the number of vOTUs indicated that the number of families and genera discovered was close to saturation (Fig. 1e). This was not the case when looking at the species level, mainly due to the presence of rare vOTUs with single member (n = 62,537) (Supplementary Fig. 1a). When only considering 19,604 vOTUs consisting of at least two conspecific viral sequences (totaling 97,941 viral sequences), a closer saturation was achieved for each taxonomic rank (Fig. 1e).

Taken together, the established ELGV catalog revealed massive hidden viral diversity early in life, as most of the ELGV catalog could not be assigned to an existing viral family and genus. Thus, the ELGV could facilitate future investigations of the early-life human gut microbiome, likely serving as a comprehensive alignment resource to facilitate the discovery of more viruses.

### Temporal development of the early-life human gut virome

To reveal the dynamics of the human gut virome early in life, the quality-controlled reads from each metagenome were mapped back to the 82,141 vOTU representatives to calculate the relative abundance of each vOTU in fecal samples (see "Methods"). A previous study has indicated differences in the composition of gut virome from VLPs-enriched or bulk metagenomes, as the former captures infecting viruses or integrated prophages while the latter targets free or active viruses[16]. We, therefore, analyzed VLPs-enriched or bulk metagenomes separately thereafter to comprehensively reveal properties of the early-life human gut virome by different sequencing approaches.

As expected, the mapping rate from VLPs-enriched metagenomes was higher (two-sided Wilcoxon test, $P < 2.2e\text{-}16$) than that of bulk metagenomes (n = 1865, median = 13.8%, IQR = 3.99–32.4% for VLPs vs. n = 6265, median = 2.14%, IQR = 1.50–2.91% for bulk; Supplementary Fig. 2a). Given the potential influence of the infant age and sequencing depth on the resulting gut virome composition, we further limited the

infant age covered by both VLPs-enriched and bulk metagenomes (i.e., from birth to two years old). We did observe that the number of vOTUs per million sequenced reads after VLPs enrichments (median = 14.5, IQR = 3.41–43.1) was higher (two-sided Wilcoxon test, $P < 2.2e\text{-}16$) than that of bulk (median = 4.94, IQR = 2.70–9.37) (Supplementary Fig. 2b).

A total of 28,531 and 64,934 vOTUs were respectively detected in 1682 VLPs-enriched and 6205 bulk metagenomes, with a median relative abundance of 218 and 298 reads per kilobase per million mapped reads (RPKM) (IQR = 99–564 RPKM for VLPs and IQR = 111–854 RPKM for bulk), which were used for subsequent analyses. We found that 11,458 vOTUs were recovered by both approaches, indicating that 40.2% of vOTUs from VLPs-enriched metagenomes could be captured with bulk metagenomic sequencing (Supplementary Fig. 2c). Furthermore, based on fecal samples that were processed by both VLPs-enriched and bulk metagenomic sequencing from ref. 15, 2234 vOTUs were captured by both approaches, accounting for 41.7% of VLPs and 34.7% of bulk (Supplementary Fig. 2c). The richness of vOTUs from bulk metagenomes was found to be higher (two-sided Wilcoxon test blocked by "study", $P < 2.2e\text{-}16$) than that of VLPs-enriched metagenomes (median = 79; IQR = 50–130 for bulk; median = 32; IQR = 8–81 for VLPs). Both sequencing approaches showed great viral variability among samples as only small parts of vOTUs were populated with a prevalence ≥5% (n = 66 for VLPs and 284 for bulk) and 1% (n = 990 for VLPs and 1561 for bulk) (Supplementary Fig. 2d, e). Moreover, the gut virome profiled by VLPs exhibited higher (two-sided Wilcoxon test, $P < 2.2e\text{-}16$) variability than that of bulk based on these vOTUs at a prevalence ≥5% or 1%, which was also confirmed when analyzing the fecal samples that were processed simultaneously by both VLPs-enriched and bulk sequencing[15].

We next examined the dynamics of the virome diversity by stratifying the metagenomes into discrete time points (months 0, 1, 3, 6,

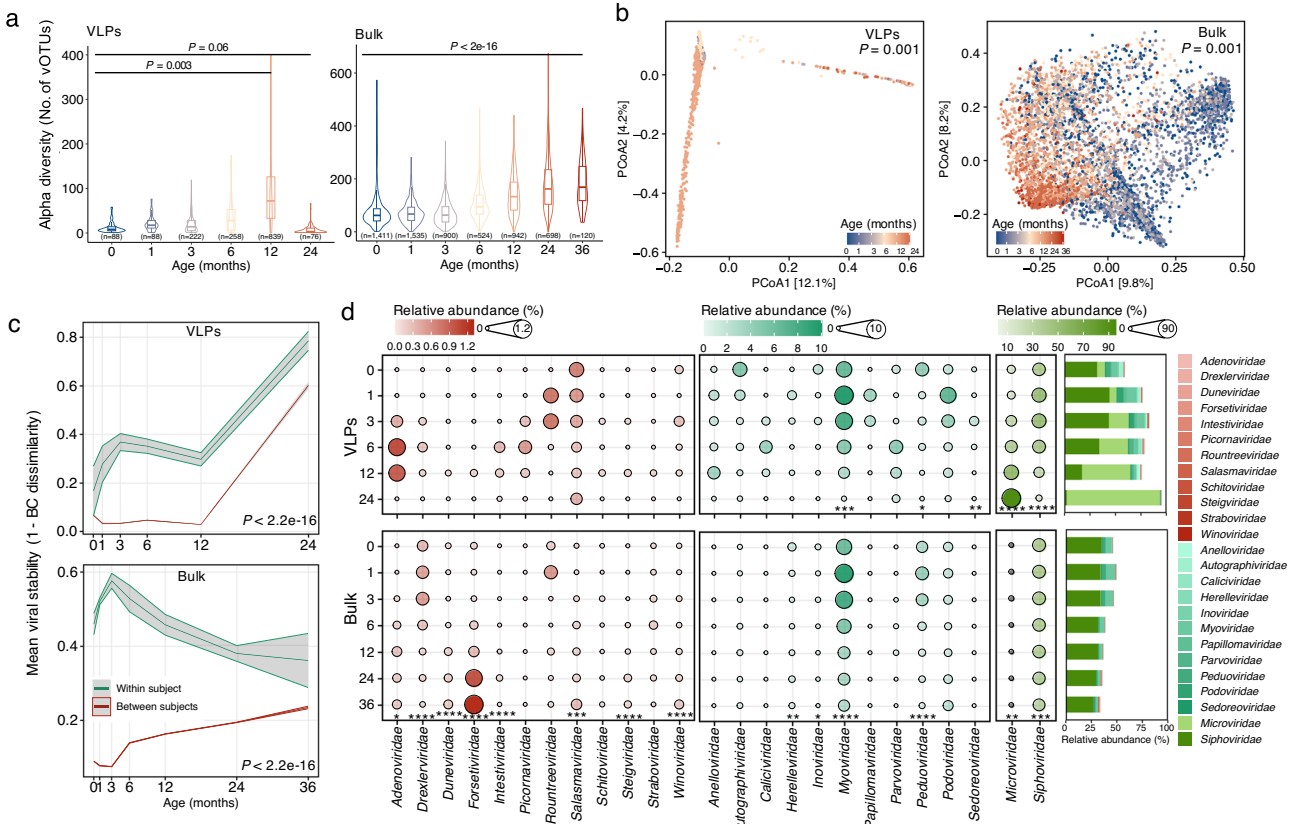

**Fig. 2 | Dynamic development of the early-life human gut virome. a** Longitudinal changes of richness (number of observed vOTUs, left for VLPs and right for bulk) with the age of infants for viral species. The boxes indicate the interquartile range (IQR), with the horizontal line as the median, the whiskers for the range of the data (up to 1.5 × IQR), and points beyond the whiskers as outliers. The *P* values were obtained with linear mixed modeling with "study" as random factor. **b** Principal coordinate analysis (PCoA) ordination of the early-life human gut virome beta diversity (*n* = 1478 for VLPs, left; *n* = 6085 for bulk, right) measured by Bray−Curtis distance with vOTU representatives ≥1% prevalence for VLPs and ≥5% prevalence for bulk. The points are colored according to the age of infants. The *P* values were obtained from PERMANOVA analysis with "study" as a block factor. **c** Viral stability within and between individuals measured by Bray−Curtis (BC) distances over the first two (VLPs, top) or three (bulk, bottom) years of life. Lines show mean BC

distance at each time point. The *P* values were obtained from two-sided Wilcoxon test blocked by "infant age", and shaded area indicates the estimated 95% confidence interval. **d** Dynamics of the relative abundance of viral families in the first two (VLPs, top) or three (bulk, bottom) years of life. Only the viral families with a prevalence >1% in VLPs-enriched metagenomes are plotted. For better visualization of the changes of each viral family, viral families are stratified into three groups based on the mean relative abundance of VLPs-enriched metagenomes at each time point (i.e., maximal mean relative abundance ≤1% (left, *n* = 12, red), maximal mean relative abundance >1% and <40% (middle, *n* = 11, light green), maximal mean relative abundance ≥40% (right, *n* = 2, dark green)). The bar plot shows the proportion of relative abundance of all 25 viral families, which are indicated in the legend on the right side. The *P* values were obtained with linear mixed modeling with "study" as random factor. \*\*\*\**P* < 0.0001, \*\*\**P* < 0.001, \*\**P* < 0.01, \**P* < 0.05.

12, 24 for VLPs and bulk, and month 36 additionally for bulk) early in life. The overall richness of the early-life gut virome dynamically increased (linear mixed modeling with "study" as random factor, *P* = 0.003 for VLPs and *P* < 2e-16 for bulk) as infants aged (Fig. 2a). We further confirmed that this increase in the alpha diversity of virome did not correlate with sequencing depth (Supplementary Fig. 3a). When stratifying the vOTUs based on their lifestyles for both VLPs and bulk, the proportion of temperate viruses in the infant gut decreased (linear mixed modeling with "study" as random factor, *P* = 0.002 for VLPs, and *P* = 0.003 for bulk), while the proportion of lytic viruses increased with infant age, which were also observed only with complete and high-quality vOTU representatives (Supplementary Fig. 3b, c). The Bray−Curtis distances based on the relative abundance of vOTU representatives indicated a strong longitudinal shift of early-life human gut virome with infant age (PERMANOVA, 1000 permutations, with "study" as a block factor, *P* = 0.001 for VLPs or bulk; Fig. 2b). We further investigated the individuality and stability of the viral profiles within and between subjects based on Bray−Curtis distances over time. The similarities within subjects from either VLPs-enriched or bulk metagenomes were higher (two-sided Wilcoxon test blocked by "infant age", *P* < 2.2e-16) than that between the subjects throughout

time, particularly in the first year of life (Fig. 2c). Notably, variations in the early-life gut virome between subjects gradually decreased as infants aged, which was indeed consistent with other members of gut microbiome, e.g., the bacterial community early in life[25].

To characterize the composition of the early-life human gut viral community, we summed the relative abundance of individual vOTU representatives in the same family rank for each metagenome to reflect the viral temporal changes at the family level. A total of 42 viral families from 1682 VLPs-enriched metagenomes with ≥1 vOTU detected accounted for 87.9% of total abundance (median, IQR = 61.6−98.8%); while 37 families from bulk metagnomes accounted for a lower relative abundance (median = 41.4%, IQR = 31.3−53.6%), indicating more unclassified viruses were captured by bulk metagenomes (Supplementary Fig. 4a). Notably, nine viral families that were absent in bulk metagenomes were detected in VLPs-enriched metagenomes, including all three RNA viral families *Astroviridae*, *Caliciviridae*, and *Picornaviridae*. There were also four viral families that were detected by bulk metagenomes but missed in VLPs-enriched metagenomes, including *Anaerodiviridae*, *Mesyanzhinovviridae*, *Vilmaviridae*, and *Zierdtviridae* (Supplementary Fig. 4b). In VLPs-enriched metagenomes, the most abundant families in the

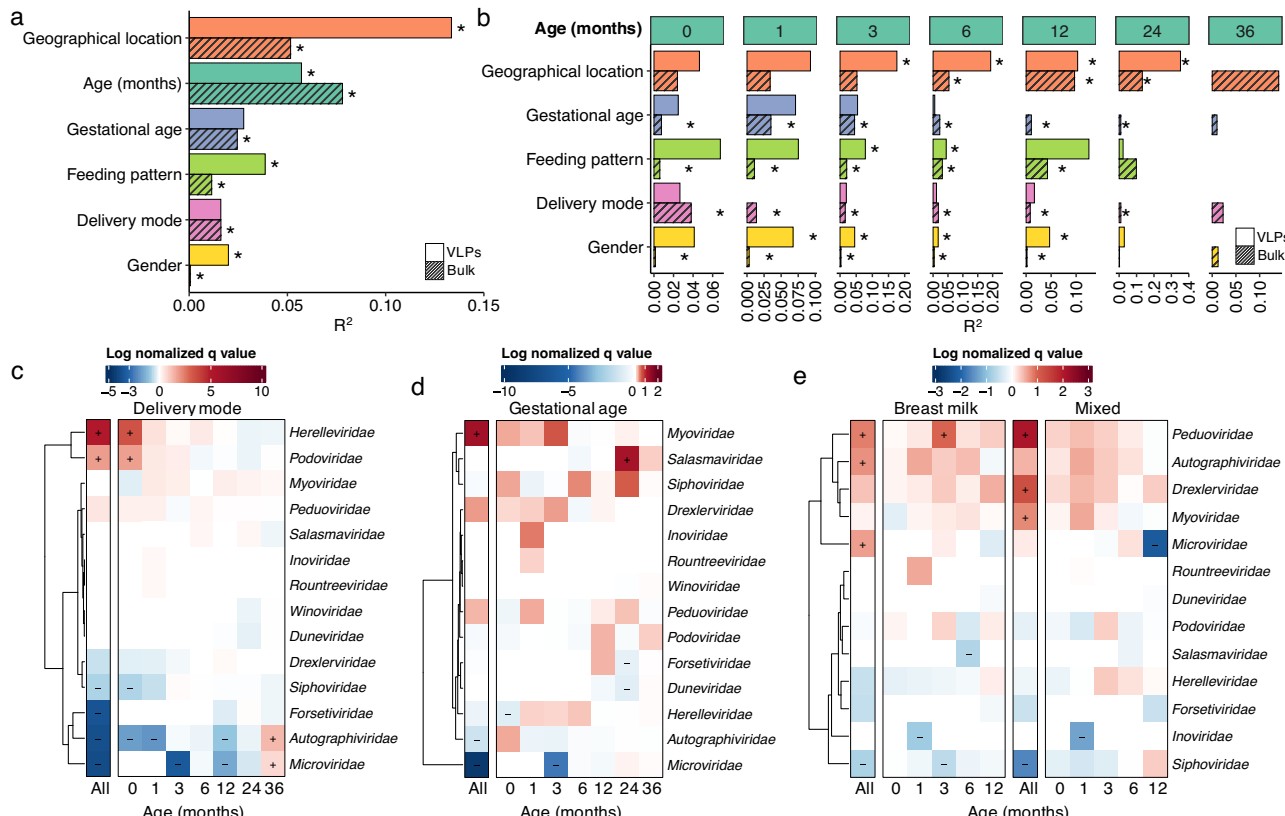

**Fig. 3 | Factors that shape the development of the early-life human gut virome.** Effect size ($R^2$) explained by clinical factors as determined by PERMANOVA (**a**) and stratified by infant age (months) (**b**) based on VLPs-enriched or bulk metagenomes. Asterisk (*) denotes the significance (FDR < 0.05) of each factor. The viral families from bulk metagenomes significantly (MaAsLin2 with "subjects" as random effect and other metadata factors as fixed effects; q < 0.25) associated with delivery mode taking vaginal delivery as reference (**c**), gestational age taking infants born full-term as reference (**d**), and feeding pattern taking non-breast milk feeding as reference (**e**), which were analyzed with all samples together or stratified by infant age (months). The sign (+) indicates positive and (−) negative associations with infants born by C-section (**c**), preterm (**d**), and fed by breast milk exclusively or partially (**e**), respectively. The stratified analysis of feeding pattern for infants at months 24 and 36 was not available due to insufficient factor levels. The color gradient indicates the strength of the association calculated by $-\log_{10}(q \text{ value}) \times \text{sign(coeff)}$.

first two years of life included *Microviridae*, *Siphoviridae*, and *Myoviridae*, however, *Microviridae* had a lower (two-sided Wilcoxon test, P < 0.05) relative abundance in bulk metagenomes that were dominated by *Siphoviridae*, *Myoviridae*, and *Peduoviridae* (Supplementary Data 4). When partitioning the fecal samples into discrete time points, we observed the dynamic succession of each individual family over time, with 7 and 21 families changing significantly from VLPs-enriched and bulk metagenomes, respectively (linear mixed modeling with "study" as random factor, P < 0.05; Supplementary Data 4). Out of 42 viral families with a prevalence >1%, 25 families accounted for >99% of viral abundance in 1608 VLPs-enriched metagenomes. Of these, five families with significant changes and only *Microviridae* increased in abundance as infants aged (Fig. 2d). When examining these 25 viral families in bulk metagenomes, 21 families were detected and accounted for >99% of viral abundance in 5990 bulk metagenomes. Of these, 14 families had a statistically significant difference (P < 0.05) in their abundance as infants aged, with half of the families increased, such as *Adenoviridae*, *Duneviridae*, and *Forsetiviridae*; and the other half decreased, such as *Siphoviridae*, *Myoviridae*, *Microviridae*, and *Peduoviridae*. Furthermore, the mean relative abundance of *Forsetiviridae* was sparse with <0.3% in the first 12 months, but steadily increased to 0.97 % and 1.24% at months 24 and 36, respectively. Of note, we also found that some families reached the peak in relative abundance at months 1 or 3 (i.e., *Myoviridae*, *Peduoviridae*, *Rountreeviridae*, *Siphoviridae*) and afterward decreased gradually, showing that certain viral taxa did not change uniformly early in life (Fig. 2d).

Regarding how the proportion of viruses with different lifestyles changed early in life from 5 or 14 altered viral families in either VLPs-enriched or bulk metagenomes, we found that there was a significant (linear regression, P < 0.05) decrease in the proportion of temperate viruses from two viral families, i.e., *Peduoviridae* and *Siphoviridae* in both approaches. In contrast, the proportion of temperate viruses from *Microviridae* in both approaches, and ten families such as *Duneviridae* and *Salasmaviridae* increased in bulk metagenomes (linear regression, P < 0.05) over time (Supplementary Fig. 4c).

**Factors shaping the development of early-life human gut virome**
To determine the extent to which clinical factors affect the early-life human gut virome, we gathered all metadata available from the included studies, including delivery mode, gestational age at birth, feeding pattern at sampling, and geographical location (i.e., categorizing by country where the fecal samples were collected). Although these factors have been extensively linked to the development of the early-life human gut bacteriome[25,35–37], their relationship with the overall development of the early-life human gut virome, which factor plays a greater role in shaping the gut virome composition, and viral members associated with each factor are poorly understood.

We thus firstly performed both the PERMANOVA and MANTEL analyses based on Bray–Curtis distances to estimate the effect size and correlation coefficient, respectively, of all metadata factors. Both analyses revealed a significant (FDR < 0.05, 1000 permutations, with "study" as a block factor for PERMANOVA) contribution and correlation between the early-life human gut virome development and all

investigated factors except for influences of delivery mode and gestational age on the gut virome profiled by VLPs (Fig. 3a for PER-MANOVA analysis and Supplementary Fig. 5a for MANTEL analysis). Among them, infant age and geographical location at sampling accounted for the highest viral taxonomic variation in both approaches (5.7% and 13.3% for VLPs; 7.8% and 5.2% for bulk; Fig. 3a), which was comparable to the similar analysis conducted for the gut bacterial development early in life[35]. Notably, we found that gestational age, delivery mode, and feeding pattern at sampling had systematic effects on the overall composition of the viral community based on bulk metagenomes with 2.4%, 1.6%, and 1.1% of variance, respectively, reflecting the sequence of factors that the newborn was exposed to at, or soon after, birth. Notably, gender was also observed with significant influences on the structure of gut virome early in life identified by both approaches.

When stratifying the infant age into discrete windows, we observed a time-dependent effect size of these factors along with the infant age (Fig. 3b), which was similar to what was previously observed for the early-life human gut bacteriome[36]. In the first year of life, delivery mode, gestational age, and feeding pattern at sampling exhibited significance (FDR < 0.05) and greater effect size on the development of the early-life human gut virome, but their predominance was changed based on bulk metagenomes. At birth (month 0), delivery mode explained the greatest amount of variance (3.8%), but its influence gradually decreased until month 24. Gestational age predominated the variance at month 1 (3.5%) and month 3 (4.5%). The effect size of feeding pattern at sampling (i.e., exclusive, partial or without breastfeeding) increased along with the increased duration of feeding, and exhibited comparable effect size as delivery mode and gestational age at month 3, and a larger effect than delivery mode and gestational age at month 12 when half of fecal samples analyzed at this timepoint were collected from breastfed infants, which was also ascertained based on VLPs-enriched metagenomes. Meanwhile, the geographical location was strongly associated with the early-life human gut virome from month 6 for both VLPs-enriched and bulk metagenomes (19.4% for VLPs and 5.3% for bulk at month 6; 35.2% for VLPs and 13.3% for bulk at month 24), indicating the role of environmental factors in determining the gut virome development later in life. These findings demonstrated that while multiple factors could influence the development of early-life human gut virome, the dominance of factors with strong effects changed along with the infant age.

We further found that infants born vaginally had higher (two-sided Wilcoxon test blocked by "study", $P = 0.004$) richness (i.e., number of vOTUs) than infants born by C-section based on bulk metagenomes, and infants fed exclusively by breast milk had lower number of vOTUs than those fed partially (two-sided Wilcoxon test blocked by "study", $P < 0.05$) or without breast milk (two-sided Wilcoxon test blocked by "study", $P < 2.6e-7$) for both VLPs-enriched and bulk metagenomes. We did not observe significant differences (two-sided Wilcoxon test blocked by "study", $P > 0.05$) between infants born full-term or preterm for both approaches (Supplementary Fig. 5b, c).

To clarify how the specific viruses in early life are associated with these three factors, we employed a linear mixed modeling as implemented in MaAsLin2[38] taking "subjects" as a random effect based on the relative abundance of vOTUs at the family level (see "Methods"). We did not observe any families significantly ($q < 0.25$) associated with delivery mode, gestational age, and feeding pattern at sampling based on VLPs-enriched metagenomes, while some significant families were identified from bulk metagenomes (Fig. 3c–e). More specifically, bulk metagenomes from infants born by C-section (taking vaginal delivery as reference in MaAsLin2 model) were enriched ($q < 0.25$) with the viral families *Herelleviridae* and *Podoviridae* although they only accounted for an average of 1.39% in relative abundance of the early-life human gut virome. In contrast, infants born vaginally harbored higher ($q < 0.25$) abundance of *Microviridae, Autographiviridae, Forsetiviridae,*

and *Siphoviridae*, and these families comprised an average of 39.1% in relative abundance (Fig. 3c). Notably, these differential commensal viral families were mainly observed at month 0 and 1, and some taxa persisted into later life. Regarding the influence of gestational age (taking full-term born infants as reference in MaAsLin2 model) on the individual viral families, we found *Myoviridae* was enriched in infants born preterm with an average of 5.95% in relative abundance; while the families *Microviridae* and *Autographiviridae* were depleted (Fig. 3d). We also compared exclusive or partial breastfeeding to non-breast milk feeding (as reference in MaAsLin2 model), and found that exclusive and partial breastfeeding enriched ($q < 0.25$) for *Peduoviridae, Autographiviridae, Drexlerviridae, Myoviridae,* and *Microviridae*, but decreased *Siphoviridae* abundance in infants (Fig. 3e). There was over 40% of vOTUs without an assignment for the viral family, we thus wondered if a finer association existed between all the vOTUs and clinical factors irrespective of their taxonomic affiliation. We found three (all positive) and 27 vOTUs (14 positive and 13 negative) that were significantly ($q < 0.25$) associated with delivery mode and feeding pattern at sampling, respectively, based on VLPs-enriched metagenomes (Supplementary Data 5). In cases of bulk metagenomes, apart from vOTUs belonging to the viral families observed above, additional vOTUs that failed to be assigned taxonomically were influenced by delivery mode, gestational age, and feeding pattern at sampling (Supplementary Fig. 6; Supplementary Data 5). For example, 200 vOTUs were associated ($q < 0.25$) with delivery mode (93 positive and 107 negative) when analyzing all fecal samples together, and 143 of 200 were not assigned at the family level, such as vOTU_30363, vOTU_55926, and vOTU_54931 being positively associated with C-section, and vOTU_36740, vOTU_18175, and vOTU_52214 being negatively associated with C-section (Supplementary Fig. 6a).

## Host prediction and close interactions between the gut virome and bacteriome early in life

We next sought to link each vOTU to its potential host by extracting the Clustered Regularly Interspaced Short Palindromic Repeats spacer sequences (CRISPR spacers) from the 32,277 early-life human gut microbial genomes (i.e., ELGG catalog) that were assembled from the 6122 bulk metagenomes[39], and then compared CRISPR spacers to vOTU representatives to determine the host-virus connections.

We identified bacterial hosts for 18% of 82,141 vOTUs ($n = 14,684$), and the most common phylum host was Firmicutes/_A/_C ($n = 10,175$), followed by Actinobacteriota ($n = 2688$), Bacteroidota ($n = 1256$), Proteobacteria ($n = 590$), and Verrucomicrobiota ($n = 191$) (Fig. 4a). The most common bacterial family host included *Lachnospiraceae* ($n = 4029$), *Bifidobacteriaceae* ($n = 2569$), *Veillonellaceae* ($n = 1410$), *Clostridiaceae* ($n = 1099$), and *Bacteroidaceae* ($n = 903$), all representing the most abundant bacterial families detected by MetaPhlAn 4[40] from the 6265 metagenomes early in life (Supplementary Fig. 7a). Of the vOTUs whose hosts were predicted at the genus level, most of them (91.1%; $n = 13,378$) were hosted by only one genus, whereas the other vOTUs (8.9%; $n = 1306$) were predicted to infect multiple genera (Supplementary Fig. 7b). The early-life human gut bacteriome is known to differ from that of adults, particularly the *Bifidobacterium* genus with higher relative abundance and prevalence in infants[23]. As expected, we observed that the majority of early-life human gut vOTUs were hosted by *Bifidobacterium* ($n = 2565$), predominantly including *Bifidobacterium longum* ($n = 1198$), *Bifidobacterium pseudocatenulatum* ($n = 1197$), *Bifidobacterium adolescentis* ($n = 1064$), and *Bifidobacterium kashiwanohense* ($n = 989$) (Fig. 4a). Of note, 120 vOTUs were predicted to be hosted by *Escherichia coli*, the species represented by the largest number of genomes in the ELGG catalog.

Given the impact of the gut virome diversity on the gut bacteriome and vice-versa in adults[11,41], we examined the temporal correlation between the composition of early-life human gut virome and bacteriome, which thus far is not clearly understood. To address this,

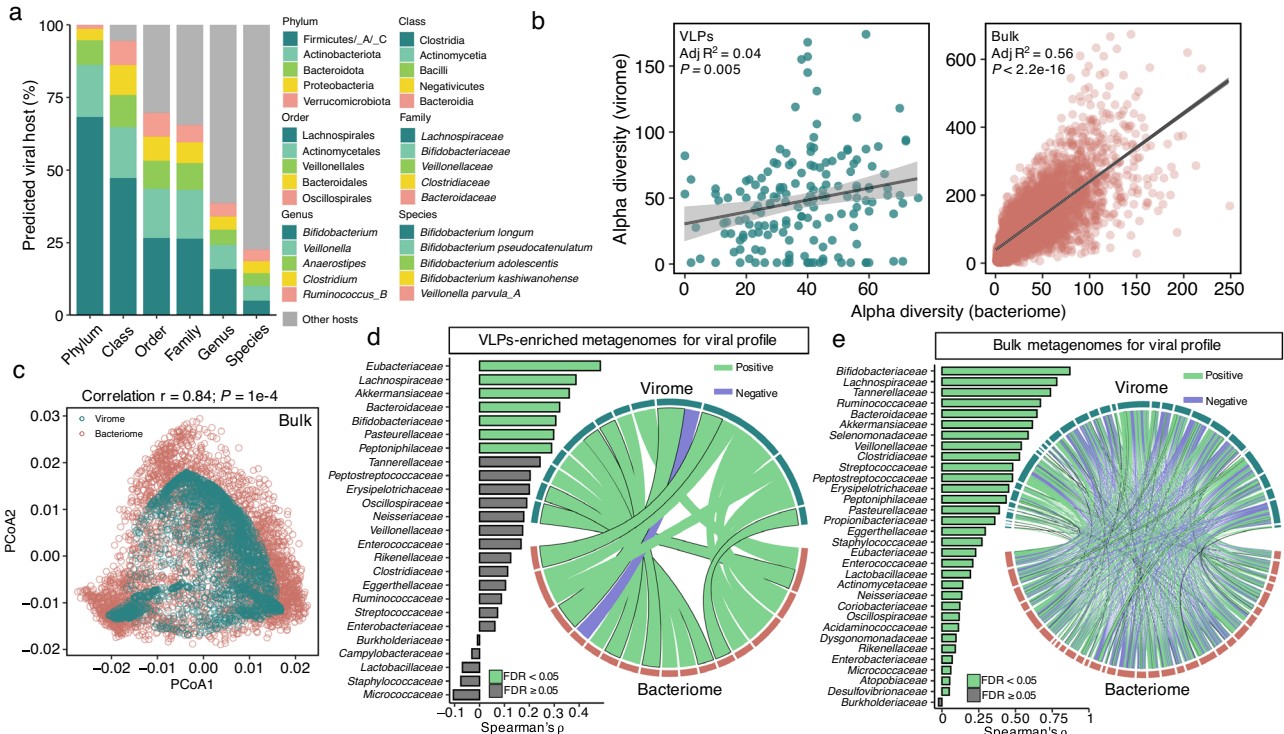

**Fig. 4 | Close interactions between the early-life human gut virome and bacteriome. a** The proportion of the top five predicted hosts at various taxonomic ranks. **b** Comparisons of alpha diversity (richness, left for VLPs, and right for bulk) between the early-life human gut virome and bacteriome at the species level. The $P$ values were obtained with linear regression, and shaded area indicates the estimated 95% confidence interval. **c** Procrustes analysis of association between the virome and its predicted bacterial host across 32 families of bulk metagenomes ($n = 3398$). Correlations between individual families between the virome (the half

dark green circle, top) and the predicted bacterial host (the half-red circle, bottom) for VLPs-enriched (**d**) and (**e**) bulk metagenomes. The positive significant correlations (band in circle) were in light green, and negative in purple (FDR < 0.05). The width of the band corresponds to the value of correlation coefficient (Spearman's $\rho$). The band colored with black border indicates the correlation between virome and its predicted bacterial host, and their width also corresponds to the bar plot on the left with Spearman's $\rho$ for each bacterial host at the family level.

we focused on the bacteriome of 141 fecal samples that were simultaneously processed by using both VLPs-enriched and bulk metagenomic sequencing from ref. 15. and 6066 fecal samples that were only subjected to bulk metagenomic sequencing to compare with the corresponding virome in each sample according to different sequencing approaches. The alpha diversities of gut virome and bacteriome (i.e., richness) at the species level showed significant correlations (linear regression, adjusted $R^2 = 0.04$ and $P = 0.005$ for VLPs, and adjusted $R^2 = 0.56$ and $P < 2.2e-16$ for bulk; Fig. 4b), which was expected for a lower correlation coefficient for VLPs than that of bulk due to differences in sequencing approaches. This significant correlation between the gut virome and bacteriome profile early in life was also observed for the beta diversity based on Bray−Curtis distances (linear regression, $P < 1.6e-6$ for VLPs and $P < 2.2e-16$ for bulk; Supplementary Fig. 7c).

To further explore specific bacteria-virus associations, we compared the relative abundance of each virus and its predicted bacterial host at the family level given the confident resolution of the viral family assignment. A total of 25 families detected by both viral host prediction and MetaPhlAn 4 from 120 fecal samples that were sequenced by both approaches were used for this part of analyses. Meanwhile, 32 families from 3398 fecal samples that were solely sequenced by bulk were used for comparisons. We first leveraged the Procrustes analysis[42] to compare the overall distribution of the gut virome and bacteriome based on the Bray−Curtis distances. We found a significant correlation with coefficient $r = 0.84$ ($P = 1e-04$) for bulk metagenomes (Fig. 4c), but not for VLPs-enriched metagenomes ($P = 0.58$). Regarding the correlations between individual families, a total of 19 (18 positive and 1 negative) and 545 (311 positive and 234 negative) significant

correlations (Spearman correlation, FDR < 0.05) between the gut virome and bacteriome were observed for VLPs-enriched and bulk metagenomes, respectively (Fig. 4d, e). Notably, 7 and 31 viral families showed significant positive correlations in relative abundance with their predicted bacterial hosts for VLPs-enriched and bulk metagenomes, respectively (Fig. 4d, e). Among them, we found that five viral families from VLPs-enriched metagenomes (except *Lachnospiraceae* and *Bacteroidaceae*) and 29 viral families from bulk metagenomes (except *Ruminococcaceae* and *Tannerellaceae*) showed the strongest correlation with their predicted bacterial host than other bacterial hosts (ranging from Spearman's $\rho = 0.48$ for *Eubacteriaceae* to 0.29 for *Peptoniphilaceae* for VLPs; from 0.87 for *Bifidobacteriaceae* to 0.049 for *Desulfovibrionaceae* for bulk).

We further built the co-occurrence networks at the species level for VLPs-enriched and bulk metagenomes, involving 1031 vOTUs and 356 bacterial species for VLPs with ≥1% prevalence, and 283 vOTUs and 159 bacterial species with ≥5% prevalence, respectively. The resulting network from VLPs-enriched metagenomes consisted of 996 nodes and 4301 edges with significant positive correlations (Spearman correlation, Spearman's $\rho \geq 0.6$, FDR < 0.05). Among them, 89% ($n = 887$) of nodes were from vOTUs, and 6.6% ($n = 285$) of edges were connected by vOTUs and bacterial species (Supplementary Fig. 8a). The bacterial species with the highest number of connections with the virome included *Bacteroides timonensis*, *Dorea formicigenerans*, and *Blautia schinkii*, and all edges from 44 bacterial species and 693 vOTUs were connected with the virome (Supplementary Fig. 8b, c; Supplementary Data 6). With bulk metagenomes, 79% ($n = 192$ of 242 in total) of nodes from the network were from vOTUs, and 23% ($n = 102$ of 436 in total) of edges were connected by vOTUs and bacterial species

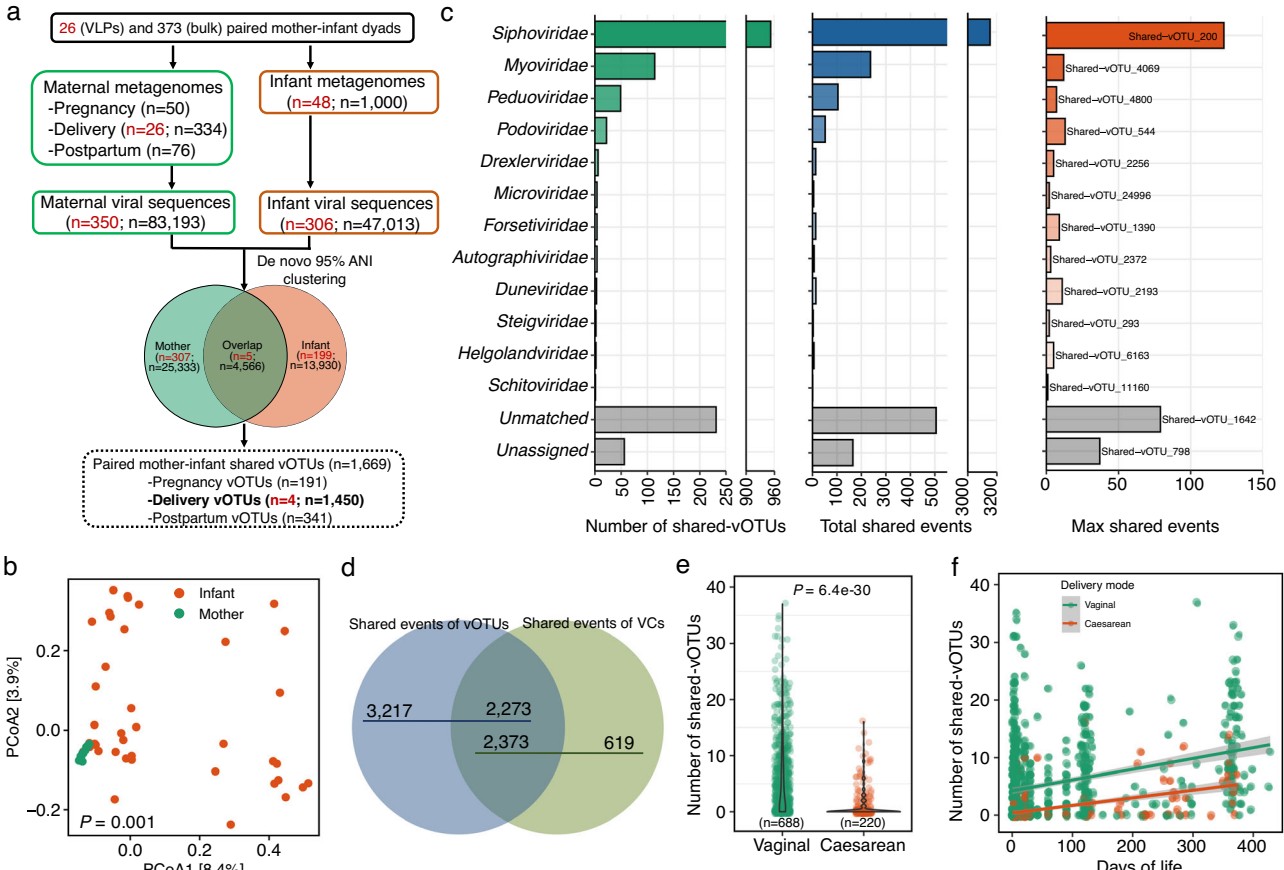

**Fig. 5 | The status of viruses shared by the paired mother-infant dyads. a** The pipeline to compare the human gut virome between mother-infant dyads from VLPs-enriched (number in red) and bulk (number in black) metagenomes. **b** Principal coordinate analysis (PCoA) ordination of the gut virome of mothers and infants measured by Bray–Curtis distance based on the relative abundance of each vOTU in VLPs-enriched metagenomes. The *P* value was obtained from PERMANOVA analysis. **c** The summary of mother-infant shared viral families, including the number of shared vOTUs and shared events, as well as the vOTUs with the maximal number of shared events within each shared viral family. **d** Comparisons between mother-infant shared vOTUs and viral clusters (VCs). Influence of delivery mode on the mother-infant shared vOTUs (**e**), and stratified by infant age (days) (**f**). The *P* values in (**e**) was obtained with two-sided Wilcoxon test blocked by "study" and shaded area in (**f**) indicates the estimated 95% confidence interval.

(Supplementary Fig. 8d). The bacterial species with the highest number of connections with the virome included *Faecalibacterium prausnitzii*, *Anaerostipes hadrus*, and *Blautia wexlerae*, and all edges from 30 bacterial species such as *Phocaeicola vulgatus*, *B. longum*, and *Staphylococcus epidermidis* were connected with the virome (Supplementary Fig. 8e). vOTU_64450 and vOTU_34897 had the highest number of connections with 19 and 18 edges, respectively, followed by vOTU_69856, vOTU_28728, and vOTU_36740, while vOTU_46177 had the highest number of connections with bacterial species with six edges (Supplementary Fig. 8f). Overall, these results highlight the close interconnection and frequent co-occurrence between specific viral and bacterial taxa early in life.

## A set of viruses shared by paired mother–infant dyads

To fully decipher the shared and unique properties of mother-infant virome particularly early in life, we analyzed all fecal samples from mothers whose infants were included in the reconstruction of the ELGV catalog. As a result, 373 paired mother-infant dyads including 460 maternal fecal bulk metagenomes (covering pregnancy, delivery, and postpartum) and 1000 infant fecal bulk metagenomes (ranging from birth to the first two years of life) were analyzed (Supplementary Data 7). Additionally, we also collected the VLPs-enriched metagenomes from 26 paired mother-infant dyads from ref. 27, including 26 maternal and 48 infant samples (Supplementary Data 7). All the maternal gut metagenomes were assembled, and the viral sequences

were predicted using the same approach applied for the infant metagenomes, which resulted in qualified 83,543 maternal viral sequences for bulk and VLPs-enriched metagenomes. Based on CheckV estimation, the median length of maternal viral sequences was 6083 bp (IQR = 4025–12,458 bp), and 4134 viral sequences were categorized to be complete or high-quality (>90% completeness), 6250 viral sequences were estimated to be medium-quality (50–90% completeness), 73,107 viral sequences were low-quality genomes (<50% completeness), and 52 viral sequences as "not-determined" (Supplementary Fig. 9a).

Based on VLPs-enriched metagenomes, we clustered 656 qualified mother-infant viral sequences into 511 vOTUs at 95% ANI over 85% AF of the shorter sequence[33]. Among them, 199 and 307 vOTUs were exclusively represented by viral sequences from mothers and infants, respectively, and only five vOTUs were shared by mothers and their paired or unpaired infants (Fig. 5a). After excluding vOTUs exclusively containing viral sequences from the unpaired mother-infant dyads, four vOTUs were shared by two paired mother-infant dyads (hereafter referred to as shared-vOTUs; Fig. 5a), belonging to *Siphoviridae* (*n* = 3) and *Podoviridae* (*n* = 1). Apart from the assembly-based approach, we further mapped the quality-controlled reads from mothers and infants to ELGV representatives to calculate the relative abundance of each vOTU representative. We found that maternal gut viromes differed from infant gut viromes based on Bray–Curtis distances (PERMANOVA, 1000 permutations, *P* = 0.001) but the average richness between

mothers and infants was comparable (two-sided Wilcoxon test, $P = 0.84$), which was consistent with reports from the original publication[27] (Fig. 5b). Only two paired mother-infant dyads were found to share vOTUs from the reads-based mapping, and one pair of them (C047) was also observed from the assembly-based approach.

When conducting the analyses on bulk metagenomes, all mother-infant viral sequences ($n = 130,206$) were clustered into 43,829 vOTUs. 4566 vOTUs were found to be shared by mothers and their paired or unpaired infants, and 1669 vOTUs of them were shared-vOTUs, including 5490 shared events (defined as the frequency of shared-vOTUs) from 273 paired mother–infant dyads involving 329 mother and 611 infant metagenomes (Fig. 5a; Supplementary Data 8). When clustering the mother-infant viral sequences at a higher level (100% ANI over 100% AF) implying the identical strain (referred to as strain-vOTUs), a larger number of 111,599 strain-vOTUs were formed, and 1704 strain-vOTUs of them were shared by the paired mother-infant dyads, including 2792 shared events from 235 paired mother-infant dyads involving 280 mother and 509 infant metagenomes. After stratifying the maternal fecal samples by sampling time, i.e., pregnancy, delivery (≤7 days after birth), and postpartum (>7 days after birth), 191 and 341 shared-vOTUs were from the maternal metagenomes collected during pregnancy and postpartum, respectively. The majority of shared-vOTUs ($n = 1450$) with 4278 shared events were observed between infants and their paired mothers at delivery, correlating with the number of maternal metagenomes included ($n = 334$ from delivery, $n = 50$ from pregnancy, and $n = 75$ from postpartum), which was thus used for subsequent analyses.

When taxonomically annotating the shared-vOTUs from bulk metagenomes of infants and their mothers at delivery based on the UniProtKB and Demovir, most of shared-vOTUs were classified into *Siphoviridae* family ($n = 952$ shared-vOTUs with 3153 shared events), *Myoviridae* ($n = 114$ shared-vOTUs with 237 shared events), *Peduoviridae* ($n = 49$ shared-vOTUs with 104 shared events), and *Podoviridae* ($n = 22$ shared-vOTUs with 52 shared events), and the other viral families contained <10 shared-vOTUs. Among the viral families, shared-vOTU_200 was the most dominant viral species with 123 shared events in *Siphoviridae* family, shared-vOTU_4069 ($n = 12$ shared events) for *Myoviridae*, shared-vOTU_4800 ($n = 7$) for *Peduoviridae*, and shared-vOTU_544 ($n = 13$) for *Podoviridae* (Fig. 5c).

Additionally by employing a network-based gene-sharing approach of vConTACT2[43] with all 130,206 mother-infant viral sequences from bulk metagenomes as input, a total of 10,744 viral clusters (VCs) were generated, containing 69,857 viral sequences from mothers or infants (Supplementary Fig. 9b). Among these VCs, 1058 VCs were shared with 2992 shared events across 297 paired mother-infant dyads, of which 919 shared-VCs with 2359 shared events were observed from infants and their paired mothers at delivery. Of these shared-VCs, only 123 shared-VCs contained genomes from mother-infant shared viral sequences and viral RefSeq, which thus were assigned to viral families of *Myoviridae* ($n = 58$), *Podoviridae* ($n = 34$), *Siphoviridae* ($n = 26$), and *Microviridae* ($n = 2$), and the predominant genera of *Lambdavirus* ($n = 18$), *Peduovirus* ($n = 14$), and *Felsduovirus* ($n = 12$) based on the vConTACT2 analyses (Supplementary Fig. 9c). We then compared the shared events discovered by the shared-vOTUs and shared-VCs, and found comparable results, where 2373 shared events from 890 shared-VCs were confirmed by 902 shared-vOTUs with 2273 shared events (Fig. 5d), further confirming the presence of virome shared by mother and infant early in life.

We also aimed to determine the influence of available metadata variables including infant age, delivery mode, gestational period, feeding pattern at sampling, and gender on the shared-vOTUs between infants and their paired mothers at delivery. Due to the limited number of shared-vOTUs from VLPs-enriched metagenomes, we focused on bulk metagenomes and found that the shared-vOTUs accounted for an average of 11.5% of the total vOTUs found in infant bulk metagenomes.

We found that C-section delivery was strongly associated with a lower number of shared-vOTUs (mean = 1.17 vs. 5.76 from infants born vaginally; two-sided Wilcoxon test blocked by "study", $P = 6.4e\text{-}30$) in infants with an overall comparison (Fig. 5e) or in a longitudinal manner along with the infant age (Fig. 5f). Other factors with significant associations included infant age (Kruskal–Wallis test blocked by "study", $P = 1.0e\text{-}6$) and gestational age (two-sided Wilcoxon test blocked by "study", $P = 0.041$) (Supplementary Fig. 9d). The number of shared-vOTUs increased as infants aged (mean = 3.54 at month 0 vs. mean = 11.37 at month 24). Infants born full-term had a higher number of shared-vOTUs compared to those born preterm (mean = 4.70 vs. 0.36). No statistical differences were observed among female or male infants (two-sided Wilcoxon test blocked by "study", $P = 0.63$) and feeding pattern at sampling (Kruskal–Wallis test blocked by "study", $P = 0.28$) (Supplementary Fig. 9d).

## Identification of specific viruses abundant in the early-life human gut virome

Compared to the metagenomes that were used to generate other existing viral databases (i.e., CHVD, GPD, GVD, and MGV), 5068 metagenomes (1089 for VLPs and 3979 for bulk) distributed across 20 studies were unique to the ELGV catalog (Supplementary Data 9). In order to explore the uniqueness of the early-life human gut virome, we further clustered the ELGV representatives to other databases with viral representatives mainly compiled from the adult gut including CHVD ($n = 45,033$), GPD ($n = 142,809$), GVD ($n = 33,242$), and MGV ($n = 54,118$) and viruses from RefSeq ($n = 14,814$). We applied the same filtering criteria as the ELGV catalog to these viral databases for an accurate comparison, which resulted in a reduced number of viral representatives for each database, i.e., CHVD ($n = 40,203$), GPD ($n = 124,775$), GVD ($n = 16,379$), and MGV ($n = 46,464$) and viruses in RefSeq ($n = 8879$). Notably, among the generated 184,507 vOTUs clustered at 95% ANI and 85% AF of the shorter sequence, 56,131 vOTUs from the ELGV catalog (68.3%) did not cluster with any viral genomes from the other databases (Fig. 6a; Supplementary Data 9). The largest vOTUs overlap between ELGV and other databases were with GPD ($n = 5325$), followed by MGV ($n = 1412$), which was expected due to their large size. Surprisingly, there were only 23 vOTUs shared by all databases, indicating the specificity of each database, which however may also be attributed to differences in the algorithms for identification and filtration of viral sequences.

Whether and what viruses that are specifically present in early life remain open. Considering the high inter-individual variability in the early-life human gut virome, we primarily focused on the vOTUs if their prevalence exceeded 2% across 1682 VLPs-enriched metagenomes with a relative abundance >0.01 (1%) in at least one metagenome, while the threshold values were ten-fold increased for bulk metagenomes up to >20% and >0.10 given there was a lower inter-individual variability and larger sample size ($n = 6205$). This resulted in 407 vOTUs from VLPs-enriched metagenomes, which accounted for an average relative abundance of 33.7% (median = 21.2%; IQR = 1.77–62.3%) with a prevalence ranging from 2.02% to 20.1%, and 335 vOTUs exclusively belonged to the ELGV catalog. To check their specificity for the early-life human gut virome, we quantified their relative abundance in 521 adult VLPs-enriched metagenomes (Supplementary Data 10), and found that 111 of 139 significantly differential vOTUs (MaAsLin2 with age (infants vs. adults) as fixed effect, $q < 0.25$) were lower in adults than that of infants (an average of relative abundance of 6.58% vs. 17.4%; Fig. 6b). With bulk metagenomes, we obtained 28 vOTUs that accounted for an average relative abundance of 14.5% (median = 12.1%; IQR = 5.72–20.2%) with a prevalence ranging from 23.4% to 32.7% (Supplementary Fig. 10). Among them, 24 vOTUs belonged to the vOTUs that ELGV uniquely had, and the other four vOTUs overlapped with GPD (vOTU_24266, vOTU_49416, vOTU_18175), GVD (vOTU_46816), and MGV (vOTU_49416). When comparing with 510

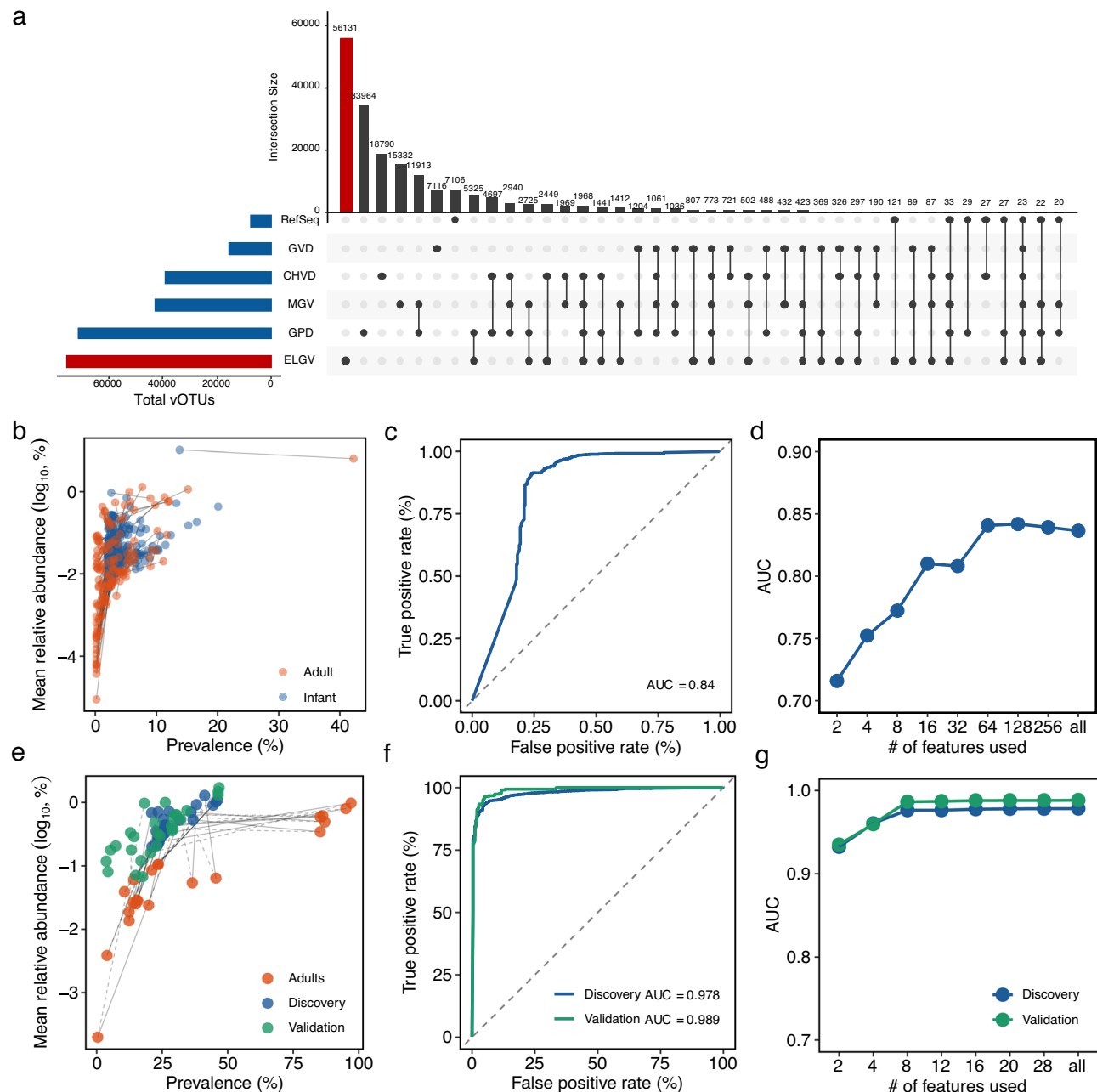

**Fig. 6 | Specific properties of viruses abundant in the early-life human gut virome. a** UpSet plot comparing ELGV ($n = 82{,}141$) against the other public human gut virome databases, including CHVD ($n = 40{,}203$), GPD ($n = 124{,}775$), GVD ($n = 16{,}379$), and MGV ($n = 46{,}464$) and viruses in RefSeq ($n = 8879$). The prevalence and mean relative abundance of 407 and 28 vOTU representatives in VLPs-enriched (**b**) or bulk (**e**) metagenomes from infants and adults. Solid lines in (**b**) link vOTUs shared by infants and adults. Solid and dotted lines in (**e**) link vOTUs shared by infants and adults from discovery and validation cohorts, respectively. Receiver operating curves with area under the curve (AUC) for prediction performances for VLPs-enriched metagenomes (**c**) and discovery and validation cohorts for bulk metagenomes (**f**). Prediction performances with an increasing number of vOTUs obtained by retraining the random forest classifier on the top-ranking features (ordered by the MeanDecreaseGINI value) identified from the random forest model trained with the full set of features from VLPs-enriched (**d**) or bulk metagenomes (**g**).

adult bulk metagenomes (Supplementary Data 10), all 28 vOTUs differed significantly (MaAsLin2, $q < 0.25$) between adults and infants, with the relative abundance of 24 being higher in infants (an average of 1.52% vs. 13.0%; Fig. 6e). Moreover, given additional bulk metagenomes of infants are publicly available, we further quantified all 28 vOTU representatives in another 302 infant bulk metagenomes from five studies that were not used to build the ELGV catalog (Supplementary Data 11), and found that all 28 vOTUs were present with comparable relative abundances, averaging 13.9% (median = 10.6%; IQR = 3.25–18.7%; Fig. 6e).

To further test whether these 407 or 28 vOTUs could be used as biomarkers to distinguish the human gut virome according to the host age, we performed predictions using a random forest classifier. With VLPs-enriched metagenomes, we randomly selected half of infant or adult metagenomes from each study and then combined for classifier training and the other half for testing, which resulted in prediction performance with area under the receiver operating curve (AUC) score of 0.84 (Fig. 6c). In cases of bulk metagenomes, we thus performed two-level (i.e., discovery and independent validation cohorts) predictions. In the discovery prediction cohort, similar to VLPs-enriched

metagenomes, half of infant or adult metagenomes for training, and the second half for testing the classifier. In the independent validation prediction cohort, the second half of adult metagenomes (i.e., that were not used for classifier training in discovery prediction) together with newly obtained 302 infant metagenomes (i.e., that were not used for classifier training and building ELGV catalog) were used for testing the classifier trained in the discovery prediction. We observed the close prediction performance with a high AUC score of 0.978 and 0.989 for discovery and validation, respectively (Fig. 6f). We then tested how many of the vOTU representatives of the early-life human gut virome were necessary to achieve the comparable predictive performance by training the classifier model with different number of top-ranking features that were chosen based on the mean decrease in GINI from the classifier trained with the full set of features. The results showed that using as few as 64 vOTUs for VLPs (Fig. 6d) and eight for bulk (i.e., vOTU_33846, vOTU_49416, vOTU_36740, vOTU_18175, vOTU_40124, vOTU_43585, vOTU_60588, and vOTU_74435; Fig. 6g, Supplementary Fig. 10) achieved AUC = 0.84 for VLPs and AUC > 0.97 in both discovery and validation cohorts for bulk. Notably, four vOTUs were overlapped by VLPs and bulk, including vOTU_49416, vOTU_36740, vOTU_18175, and vOTU_40124, leading to AUC > 0.95 in both discovery and validation cohorts for bulk but only AUC = 0.53 for VLPs. Therefore, these results highlighted that similar to the early-life human gut bacteriome, the early-life human gut virome also had specific properties and differed from that of adults.

After exploring the functional potential of these 407 or 28 vOTUs by mapping their representatives against KEGG[44], COG[45], and Pfam[46] databases, we observed that 34.6% of total genes (1655 of 4785 genes in total) for VLPs or 71.5% for bulk (123 of 172 genes in total) had a match to at least one of these databases (Supplementary Data 12). Among those matched genes, the most common functions were replication, recombination and repair, cell wall/membrane/envelope biogenesis, and transcription for both approaches, and nucleotide transport and metabolism in particular for bulk from COG annotation; and metabolic pathways (ko01100, map01100) for both approaches, DNA replication (ko03030, map03030) and homologous recombination (ko03440, map03440) for VLPs, purine metabolism (ko00230, map00230) and pyrimidine metabolism (ko00240, map00240) for bulk based on KEGG annotation. In addition, a large number of genes were homologous to capsid proteins (PF02305) and phage integrase family (PF00589) for VLPs, and proteins of glutaredoxin (PF00462), helix-turn-helix (PF01381), and ribonucleotide reductase (PF02867, PF00317, PF00268) for bulk in the Pfam database.

## Discussion

With the growing volume of metagenomic sequencing data from the human gut microbiome, the integrative analysis of combined datasets has a high chance to provide more valuable insights and resources for future studies. The existing gut viral databases generated from the pooled VLPs-enriched and bulk metagenomes greatly expand our knowledge of the gut viral composition and evolution, which however mainly focus on adults. The comprehensive metagenomic landscape of the early-life human gut virome is poorly understood, limiting the evaluation and discovery of the specific viruses early in life and their roles with pediatric diseases. In the present study, a viral catalog with 160,478 genomic sequences exclusively targeting the early life of humans has been generated from 1865 VLPs-enriched and 6265 bulk human gut metagenomes in the first three years of life, estimated to cover 82,141 species-, 11,413 genus-, and 1238 family-level vOTUs. Notably, the accumulation analysis of vOTUs indicated an approaching asymptote at the family and genus levels. By leveraging this newly established early-life human gut virome catalog, we have addressed open questions concerning the temporal composition and development trajectory of early-life human gut virome, clinical factors strongly associated with the early-life human gut virome, interactions between

early-life human gut virome and bacteriome, the status of mother-infant shared viruses, and a core set of early-life human gut viruses differing from adults, under different circumstances of sequencing approaches of VLPs-enriched or bulk metagenomes.

Numerous studies have indicated that the early-life human gut bacteriome exhibits different properties in terms of composition and functions when compared to that of adults, and these differences gradually decrease as infants grow, becoming closer to adult-like by about three years of life[23,36,39,47]. However, the maturation dynamics of the early-life human gut virome are still underexplored[48]. Since phages account for the highest proportion of viruses in the gut[34], it is speculated that the development of the early-life human gut virome is closely correlated with the dynamics of bacteriome early in life. We thus analyzed the early-life human gut virome with a large-scale infant population by the ELGV catalog created in the current study, and found that in line with the bacteriome, the richness of viral species increased in the first three years of life. In contrast, the proportion of temperate viruses gradually decreased, partially supporting the hypothesis that the composition of the early-life human gut virome is likely dominated by lysogenic induction from the pioneer bacteria[4]. We also observed that the early-life human gut virome showed great inter-individual variations that gradually decreased as infants grew, and revealed that clinical factors (i.e., delivery mode, gestational age, and feeding pattern) shifted the dominance in sequence to drive variations in the early-life human gut virome throughout time. Of note, the factors investigated here have been extensively documented to influence the bacteriome early in life, but the exact mechanism has yet to be established. We further found positive correlations between the early-life human virome and bacteriome in their diversities and abundances, which is in accordance with similar findings in adults[11]. However, the bacterial hosts of virome early in life are mainly composed of *Bifidobacterium* species, which are known as the most dominant taxa in infants. These results indicate the potential roles of the virome in shaping the bacteriome early in life, but future research is needed to explore the underlying regulatory mechanisms. A recent study carried out in vitro and in vivo revealed that the gut environment might influence the interactions between bacteria and phages by regulating bacterial gene expressions involved in functions including bacterial receptor biosynthesis and biofilm formation[49]. These findings raise the question whether transition from the aerobic/facultative anaerobic gut environment early in life to strict anaerobic conditions later in adulthood could influence the coexistence of phages and bacteria.

The shared gut microbes by mothers and their infants are well-documented with multiple lines of evidence at various taxonomic resolutions, particularly at the strain and single nucleotide polymorphism (SNP) levels[22,26,50]. These shared microbes are thought to be involved in the development and establishment of the early-life human gut microbiome and the offspring's health[26,51,52]. However, whether the maternal gut virome can also be shared with their infants and to what extent are largely unknown. Based on our findings from the largest population of mother–infant dyads, mothers and their infants did share some viral members, consistently indicated by two different cluster algorithms, i.e., ANI comparing with different clustering criteria and whole genome gene-sharing networks. However, it has to be mentioned that the lack of data on the viral profiles of other body sites (e.g., breast milk, oral cavity, and skin) and shared environments by mothers and infants hindered us from further confirming the viruses that are uniquely contributed by the maternal gut. The extent of viral sharing between mothers and infants was shown to be affected by various factors, including infant age, gestational age, and delivery mode, expanding the previous findings mainly focusing on the mother-infant shared gut bacteriome[25,26]. Ultimately, how the shared virome affects the composition and development of the early-life human gut bacteriome remains to be further investigated. Vatanen et al. reported that the mother-infant shared viruses were likely to

mediate mother-to-infant gene transfer between strains that were not vertically transmitted[24]. Moreover, interactions of these maternal microbes and their functions in infants' health are just beginning to be studied[51,52]. Given the gut bacteriome has been suggested to be a vital contributor for various human diseases[35,53], the potential roles of the viruses that infect these bacteria should be further investigated[54]. It has to be mentioned that the majority of gut viruses of infants we discovered belong to phages, and the shared status of eukaryotic viruses by mother–infant dyads has been largely overlooked. A recent study found distinct evolutionary patterns between phages and eukaryotic viruses in infants, where only the phage composition was found to become increasingly similar to their mothers as the infant aged[28]. Therefore, deeper insights into the dynamics and shared status of eukaryotic viruses are needed.

Although there is a high inter-individual variation of the gut virome in both infants and adults[55], we found that the early-life human gut virome contained a number of viral species with higher prevalence and abundance compared to adults, highlighting various viral taxa that are depleted throughout life. Importantly, we developed machine learning models with these viruses that were able to distinguish infants from adults with an average performance of >0.80 AUC for VLPs-enriched metagenomes and >0.97 for bulk metagenomes. Meanwhile, near-maximal accuracies were achieved with as few as 64 and 8 viral species, highlighting their age-dependent specificity. Further, exploring the utility of the gut virome as biomarkers of disease is a promising prospect to develop new diagnostic and treatment approaches. For example, Kaelin and colleagues identified viral signatures from the early-life human gut virome that preceded NEC onset in preterm infants[5], however whether these viruses participated in the pathogenesis of NEC remains unclear. It thus will be of interest to uncover more associations of the early-life human gut virome with various pediatric diseases and then mechanisms in detail by using our newly established early-life human gut virome catalog.

Currently, two main approaches exist for studying the viral profile within a microbiome: VLPs-enriched metagenomic sequencing and bulk metagenomic sequencing. Both approaches have pros and cons. For instance, VLPs-enriched metagenomes may skew viral profiles and abundances due to incomplete removal of cellular organisms and exclude large viruses by size filtration and whole-genome amplification. On the other hand, the use of bulk metagenomes may miss low-abundance viruses, especially if samples are not sequenced at sufficient depth[18,56]. By conducting similar analyses from VLPs-enriched and bulk metagenomes separately, we observed comparable results in profiling of the human gut virome early in life, such as the increased alpha diversity and decreased inter-individual variability. There was a proportion of ~40% viruses that overlapped by VLPs-enriched and bulk metagenomes sequenced from either the same fecal samples or the combined metagenomes, which is higher than the proportion (~10%) previously estimated in adults[16]. We propose that this inconsistency may be attributed to the host age, as a relatively simple gut virome was found early in life and gradually becomes diverse later in life[4]. Additionally, the RNA viral families were as expected only detected in VLPs-enriched metagenomes. Viral families from VLPs-enriched metagenomes were mainly dominated by *Microviridae* and *Siphoviridae*, which is consistent with the findings from Walters et al.[28]; whereas the abundance of *Microviridae* was much lower in bulk metagenomes. Moreover, VLPs-enriched metagenomes revealed larger viral variability between samples, which may partly explain the lower association of the gut virome with clinical factors (e.g., delivery mode and gestational age) and an increased number of predictive features that were necessary to distinguish the gut virome from infants and adults.

Although the current study has advanced our understanding of the gut virome in early life both in depth and breadth, there are several limitations that need to be addressed in the future. First, the majority of gut metagenomes are still from bulk metagenomic sequencing, which means some low-abundant viruses present in the infant gut may be underrepresented in the ELGV catalog. Furthermore, the application of VLPs enrichment procedures may be necessary for the discovery of active viruses that do not integrate into the host genome. Additionally, although RNA viruses represent a minor fraction in the infant gut virome[15,21,57], they have been largely excluded from the ELGV catalog due to limited metatranscriptomic sequencing data available from the infant gut. With an increasing volume of viral sequencing data from the infant gut microbiome, an expanded integrated and unified viral catalog including DNA and RNA viruses from early life should be then generated.

In summary, the established early-life human gut virome catalog including the largest number of infant fecal samples in this study comprehensively expands our knowledge of the viral diversity present in the first few years of life, providing more insights into its diversity and the factors that shape the human gut virome composition throughout time. Beyond this study, having the ELGV catalog can aid infant gut virome research serving as a resource, and facilitate to uncover the hidden associations between the gut virome and infant health.

## Methods

### Publicly available early-life human gut metagenomic datasets and assembly

PubMed with terms "(infant) AND ((gut) OR (enteric) OR (intestine)) AND (virome)" were combined to search studies that included fecal virome sequencing data from infants (up to October 2023). Datasets were subsequently manually curated to remove studies that did not correctly match the relevant metadata or did not have any sequencing data available. After this selection process, nine studies including 1865 VLPs-enriched metagenomes were retrieved. In addition, 143 bulk metagenomes were added from one of the nine studies (ref. [15]) as these samples were processed by using both VLPs and bulk metagenomic sequencing. We further used all bulk metagenomes ($n = 6122$) that were used to build the catalog of early-life human gut genomes (i.e., the ELGG catalog, ref. [39]) for mining the viral sequences (Supplementary Data 1). In total, 8130 fecal metagenomes (1865 VLPs-enriched and 6265 bulk metagenomes) were globally distributed among 15 countries across five continents. All collected metagenomes were quality-controlled and decontaminated of human genomic DNA (hg19 human reference genome) by KneadData v0.7.2 with default parameters. The quality-controlled reads ($1.36 \times 10^{11}$ paired reads, 86% of the raw sequencing reads) were assembled with MegaHIT v1.1.3[58] (default parameters except option "-min-contig-len 1000"), generating 32,647,394 contigs with a total length of $1.80 \times 10^{11}$ bp and an average N50 of 36,915 bp.

### Viral sequence prediction from early-life human gut metagenomes

To maximize the discovery potential of new putative viral sequences from the early-life human gut metagenomes, three viral detection tools (i.e., VirFinder v1.1[29], VIBRANT v1.2.1[30], and VirSorter2 v2.2.3[31]) relying on different algorithms for viral detection were run on the contigs from each metagenome. VirFinder discriminates viral sequences based on the different $k$-mer frequency signatures between viruses and hosts using machine learning models trained with previously known viral and host genomes. Only the putative viral sequences from VirFinder with a score >0.9 and $P < 0.01$ were kept in order to retrieve high-confidence predictions. In contrast to VirFinder, VIBRANT was designed to utilize a hybrid machine learning and protein similarity approach instead of sequence features for viruses recovery, and we ran VIBRANT with default settings except "-f nucl". VirSorter2 as the latest viral identifier by integrating a collection of customized automatic classifiers to improve the classification accuracy of all types of viruses, which was run with default settings (including --include-groups

dsDNAphage,ssDNA --min-score 0.5) with additional option "--keep-original-seq". Afterwards, all putative viral sequences from three viral identifiers were combined on a per-sample basis, generating a number of 3,375,049 viral sequences.

Thereafter, we ran CheckV v1.0.1[32] with "end_to_end" mode with default settings on all 3,375,049 viral sequences, and those ($n = 625,512$) including viral sequences and proviruses without the host contamination, which met the following criteria (1) with higher number of viral genes than host genes, (2) longer than 3 kbp, (3) the times the viral sequences is represented in the contig less than/equal to one (kmer_freq ≤ 1), and (4) without warning ">1 viral region detected" and "contig >1.5× longer than expected genome length", were kept for further dereplication by CD-HIT-EST v4.8.1[59] in two steps. The first step was to dereplicate on a per-sample basis, and afterwards, the cluster representatives were dereplicated across all fecal samples, and both steps were run with a global identity threshold of 99% (options "-c 0.99 -g 1 -M 0 -d 0 -n 10"). After the second dereplication, we obtained 160,498 viral sequences.

## Clustering of viral sequences into vOTUs
The nucleotide BLAST database of all 160,498 viral sequences was built by makeblastdb (option "-dbtype nucl") from blast v2.13.0, and the pairwise comparisons were generated by blasting all viral sequences all-against-all with blastn (option "-max_target_seqs 10,000"). Afterwards, two custom scripts (anicalc.py and aniclust.py) from the CheckV repository (https://bitbucket.org/berkeleylab/checkv/src/master/) were used to compute ANI and AF for clustering into species-level vOTUs on the basis of 95% ANI and 85% AF of the shorter sequence (options "-min_ani 95, -min_tcov 85, -min_qcov 0")[33]. The longest viral sequences were chosen as the cluster representatives of vOTUs. The identical viral strains were identified with 100% ANI over 100% AF for mother-infant shared viruses analysis.

To further cluster the species-level vOTU representatives at a genus- and family-level, we adopted the scripts developed by ref. 18 with a combination of gene sharing and AAI. Briefly, the open-reading frames (ORFs) of the nucleotide vOTU representatives were predicted by Prodigal v.2.6.3[60] with option "-p meta", and totaling 1,844,038 ORFs were then subjected to all-versus-all alignments by blastp from the DIAMOND v2.0.15[61] (option "-evalue 0.00001, -max-target-seqs 10,000"). For each pair of alignments, the shared genes (e value < 0.00001) were kept to compute the percentage of shared genes and their AAI. We considered genomes belonging to the same family if they shared ≥10% or ≥8 genes, and ≥20% AAI (option "-min_percent_shared 10, -min_num_shared 8, -min_aai 20") were kept, and a Markov Cluster Algorithm (MCL) v14-137 inflation factor of 1.2 was then applied. At the genus level, we used a threshold of ≥20% or ≥16 genes, and ≥50% AAI (option "-min_percent_shared 20, -min_num_shared 16, -min_aai 50") were kept, and a MCL inflation factor of 2.0 was then applied.

## Viral taxonomic annotation and lifestyle prediction
Taxonomy of viral sequences was annotated by mapping ORFs of vOTU representatives to the viral protein database in UniProtKB (including TrEMBL and Swiss-Prot; Release 2022_03) using blastp from the DIAMOND with options "-evalue 0.00001, -max-hsps 1, -max-target-seqs 10000". The corresponding taxonomic rank information was collected from NCBI based on the taxonomic identifiers of proteins in September of 2022. Afterwards, the family-level taxonomic annotations were assigned to viral sequences using Demovir R script (https://github.com/feargalr/Demovir) with default parameters and the updated database as described above using a voting approach to assign a taxonomy to each viral sequence.

Additionally, the resulting protein sequences of viral sequences from mother-infant dyads were used as input for vConTACT2[43] clustering with the default RefSeq prokaryotic viral database "--db ProkaryoticViralRefSeq211-Merged".

Viral ORFs of each vOTU representative were used to predict the lifestyle of viruses as "Lytic" or "Temperate" using PHACTS[62] with default settings. Ten replicate PHACTS predictions were performed.

## Determining relative abundance of vOTU representatives in metagenomes
The RPKM (reads per kilobase per million mapped reads) values were used to represent relative abundances of viral sequences in the fecal metagenomic samples estimated by CoverM v0.6.1 (https://github.com/wwood/CoverM). The viral vOTU representatives ($n = 82,141$) were used to build the mapping database with BWA v0.7.17-r1188, and then the quality-controlled reads from each sample were mapped using CoverM v0.6.1 with options "--mapper bwa-mem, --min-read-aligned-percent 95, --min-read-percent-identity 90, --min-covered-fraction 75, --methods rpkm" to keep high-quality mappings. During the mapping process, the quality-controlled reads were excluded if the percent identity was less than 90%, and the minimum coverage of each vOTU representative for the threshold of presence was set to 75%[63]. The obtained RPKM values of vOTU representatives within one metagenome were normalized by total RPKM value for the percentage calculation. The mapping rate of quality-controlled reads in each sample was also obtained from the output of CoverM.

## Viral host prediction
The early-life human gut genomes (i.e., the ELGG catalog)[39] were used to predict CRISPR spacers by combining the results from CRT and PILER-CR with default parameters using the script from ref. 18. The ELGG contains 32,277 microbial genomes reconstructed from infants in the first three years of life, representing 2,172 species. With this approach, we identified 405,859 spacers from 17,837 genomes. The predicted CRISPR spacers were then compared against viral vOTU representatives from the ELGV catalog using blastn optimized for short alignments with options "-evalue 0.0000001, -gapopen 10, -gapextend 2, -reward 1, -penalty −1, -word_size 5, -perc_identity 100, -max_target_seqs 10000".

## Comparisons to viral sequences from other databases
Viral sequences from four human virus databases, i.e., CHVD ($n = 45,033$), GPD ($n = 142,809$), GVD (n = 33,242), and MGV ($n = 54,118$) and viruses from NCBI RefSeq (release 214; $n = 14,814$) were obtained. The quality of all viral sequences was assessed by CheckV using the same criteria applied to generate the ELGV catalog. A nucleotide BLAST database of all quality-controlled viral sequences was built and pairwise comparisons were generated by an all-against-all blastn search.

## Taxonomic annotation of the bacteriome
Quality-controlled sequencing reads from bulk metagenomes were taxonomically annotated at the species level by MetaPhlAn 4 v4.0.2[40] using the default settings, and the unclassified fraction was estimated with option "--unclassified_estimation". MetaPhlAn 4 uses ~5.1 million unique clade-specific marker genes identified from ~1 million reference and metagenomic assembled genomes, including bacteria, archaea, and eukaryotes.

## Functional annotation of viral genomes
The ORFs of viral genomes were predicted using Prodigal with option "-p meta", and the genes were functionally annotated by eggNOG-mapper v2.1.7 based on database v5.0.2[64,65] with default settings. The functional annotations from KEGG[44], COG[45] and Pfam[46] were obtained from the the eggNOG-mapper results.

## Statistical analysis
*Effect size and correlation estimation* The effect size ($R^2$) and significance of clinical factors were calculated with a cross-sectional and

univariate PERMANOVA test with 1000 permutations based on Bray–Curtis distance of relative abundance (expressed as percentage, %) of vOTUs in each sample among samples using the "adonis2" function from the R package "vegan" v2.6-4[66] with "study" as a block factor. Considering the high inter-individual variability and to improve computational efficiency, Bray–Curtis distances were only calculated with 284 vOTUs with a prevalence ≥5% across 6205 bulk metagenomes and 990 vOTUs with a prevalence ≥1% across 1682 VLPs-enriched metagenomes that contained ≥1 vOTU. Metagenomes without the required metadata for the given factors were filtered before PERMANOVA analysis. Correlations between clinical factors and viral structure reflected by Bray–Curtis distances were examined with the function "mantel" from the R package "vegan" v2.6-4 by calculating the Spearman coefficient with 1000 permutations. The obtained $P$ values were corrected for multiple testing with the Benjamini–Hochberg method using a False Discovery Rate (FDR) threshold of 5%.

*MaAsLin2 analysis* Associations between viral features (the relative abundance, %) and clinical factors (including delivery mode, gestational age, feeding pattern at sampling, infant age, geographical location, and gender) were estimated using a multivariate linear mixed modeling as implemented in MaAsLin2[38]. All investigated clinical factors were treated as fixed effects and infant age was not included when fecal samples were stratified by infant age, with "subjects" as random effect. A prevalence threshold of 1% and 5% was set for viral feature selection from VLPs-enriched and bulk metagenomes, respectively, and the normalized relative abundances with total sum scaling (TSS) were then arcsin-square root-transformed (AST) before analyses. The resulting associations were considered significant if $q < 0.25$ was reported.

*Procrustes analysis* The association between the composition (the relative abundance, %) of the virome and its predicted bacterial host was analyzed using the Procrustes correlation[42]. The viral and bacterial abundance matrix at the family level were Hellinger-transformed, then Bray-Curtis distances were calculated, respectively. Principal coordinate analysis (PCoA) ordinations were generated, which were then rotated by the function "procrustes" from the R package "vegan". The symmetric correlation coefficient and $P$ value were calculated with 9999 permutations with the function "protest" from the R package "vegan".

*Correlation network analysis* Spearman's correlation coefficients ($\rho$) based on the relative abundance (%) of bacteriome and virome of each sample were obtained by the "corAndPvalue" function from R package "WGCNA" v1.71[67]. Co-occurrence networks were constructed based on the species of bacteriome and virome with a prevalence of ≥1% for VLPs-enriched metagenomes and ≥5% for bulk metagenomes. Robust correlations with Spearman's $\rho \geq 0.6$ or $\leq -0.6$ and FDR < 0.05 were used to construct networks, which have been used in the previous study[68]. The network was then built by using R package "igraph" v1.3.5[69].

*Random forest-based machine learning approach* A set of 521 adult gut VLPs-enriched metagenomes from six studies with >100,000 sequencing reads were obtained and used for classifier training and testing. An additional set of 302 infant (within one year old) and 510 adult gut bulk metagenomes with >100,000 sequencing reads were obtained from five studies, respectively, and used for the classifier discovery and validation. These metagenomes were quality-controlled using KneadData with default settings. The function randomForest from the R package "randomForest" v4.7-1.1 was applied to train the model with 1000 estimator trees[70]. Predictions and performance metrics were estimated with the "predict" function, and functions "prediction", "performance" from the R package "ROCR" v1.0-11[71]. The rank of predictive features by values of MeanDecreaseGINI was obtained by the function "importance" from R package "randomForest".

*Dynamic analysis of viral development* To quantify the dynamics of the early-life human gut virome throughout age, continuous infant age if available were stratified into seven specific age periods, which were chosen to both keep the maximal number of metagenomes for each period and also reflect the early-life human gut viral development. The six or seven age periods included for VLPs-enriched or bulk metagenomes: month 0 (0–7 days; $n = 111$, median of days (MD) = 1, IQR = 0–2 days for VLPs; $n = 1458$, median of days (MD) = 4, IQR = 2–7 days for bulk), month 1 (8–30 days; $n = 97$, MD = 21, IQR = 14.5–28 days for VLPs; $n = 1537$, MD = 21, IQR = 14–22 days for bulk), month 3 (31–90 days; $n = 276$, MD = 64, IQR = 44–90 days for VLPs; $n = 907$, MD = 53.5, IQR = 41.5–78 days for bulk), month 6 (91–180 days; $n = 346$, MD = 120, IQR = 120–128 days for VLPs; $n = 527$, MD = 122, IQR = 120–138 days for bulk), month 12 (181–360 days; $n = 841$, MD = 360, IQR = 360–360 days for VLPs; $n = 942$, MD = 285, IQR = 216–355 days for bulk), month 24 (361–720 days; $n = 76$, MD = 472, IQR = 398–524 days for VLPs; $n = 699$; MD = 510, IQR = 405–669 days), month 36 (>720 days; no sample available for VLPs; $n = 120$; MD = 858, IQR = 799–1085 days). The linear regression analyses were assessed with function "trendline_sum" from R package "ggtrendline" with option "model = line2P and Pvalue.corrected = FALSE".

*Blocked analysis with study* To account for the potential heterogeneity between studies, the significance of individuality, stability, diversity, mother-infant shared status of the gut virome between groups was examined using a blocked (by "study") two-sided Wilcoxon test for two-level comparisons and Kruskal–Wallis test for three or more-levels comparisons implemented in the R package "coin" v1.4-2[72]. The longitudinal changes of features along with the infant age were tested with linear mixed modeling with "study' as random factor. All statistical analyses have been stated in the relevant context.

All other quantification and statistical methods have been provided in the relevant text, with $P$ value or corrected $P$ value (FDR value) reported. A $P$ value or FDR < 0.05 was considered statistically significant.

**Reporting summary**
Further information on research design is available in the Nature Portfolio Reporting Summary linked to this article.

## Data availability
The ELGV catalog and representatives generated in this study have been deposited in the link https://doi.org/10.6084/m9.figshare.21901557.v2. Data supporting the findings are available within the paper and additional files. Source data are provided with this paper.

## Code availability
All the tools and scripts used for the data analysis are publicly available, and the version and parameters used have been indicated. The viral detection and classification pipelines were adapted from: https://github.com/alexmsalmeida/virsearch.

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

## Acknowledgements

This work was supported by the National Natural Science Foundation of China (82100590 and 82241036 to S.W., 81971433 and 82271749 to D.M., 81971428 to Y.Q., and 82201905 to J.Y.), the National Key Research and Development Program (2021YFC2701704 and 2021YFC2701700 to D.M.), the Department of Science and Technology of Sichuan Province (2023NSFSC0579 to S.W., 2023NSFSC0544 to S.L.), and the Fundamental Research Funds for the Central University (SCU2023D006). A.A. is supported by a Career Development Award from the Medical Research Council (MR/W016184/1).

## Author contributions

S.Z., D.M., and S.W. conceived the study. S.Z. and S.W. performed the analyses. A.A. S.L., J.Y., H.W., Y.Q., R.P.R., C.S., Z.Z., and X.N. contributed to the interpretation of the findings. S.Z. and S.W. wrote the manuscript. All authors read, edited and approved the manuscript.

## Competing interests

The authors declare no competing interests.
