## [Peer Review File · Nature Communications]

REVIEWER COMMENTS

Reviewer #1 (Remarks to the Author):

Zeng et al. present mined publicly deposited infant stool datasets to present a catalog of infant gut viruses. Using this dataset they then attempt to better describe the infant virome over time, variation by different metadata, and shared sequences with maternal gut samples available.

Major concerns:

1. The vast majority of sequence used is derived from bulk sequenced datasets rather than VLP processed samples. Therefore, the viruses detected are most likely integrated phage. This is fine if the goal is to make a catalog, but the authors then go on to perform analyses and make conclusions about the active infant virome which may be unfounded. For example, correlations between gut virome and bacteriome don't seem surprising given the metagenome (not VLP) dataset.
2. RNA viral component is not addressed here which is glossed over.

Minor concerns:

1. Specific parameter they used for some software are not clearly stated. For instance, on Line 591-593, they mentioned option `-keep-original-seq` for `virsorter2`, and what are the included groups and minimum score cutoff?
2. Line 576-577: They used KneadData for quality control and human contamination removal. Did the authors perform quality control on all the collected metagenomes in a uniform manner or if did they trimmed them individually based on the specific methods used to generate the metagenomes?
3. Line 595-599: Did the authors remove the flanking host region from prophages? It would be beneficial to clarify whether the resulting catalogue includes host sequences for the identified viral sequences
4. Figure 1B. What does the text in the bottom `# of viral sequences` supposed to show? It seems something may be missing?
5. Line 124 What was the other 0.2% of virus? Vertebratae viruses? Is this skewed due to the majority being non-VLP prep?
6. Line 128. The number of microviridae seems very small.
7. Line 156. What are the authors talking about regarding potential bias?

8. Line 162. Why did the authors choose >50% coverage and >95% identity?
9. A key paper Walters et al. Cell Host Microbe. 2023 Feb 8; 31(2): 187–198.e3. is missing particularly from discussion regarding shared infant/maternal virome.
10. Line 413. State extensive virome shared, but then states it is only 11.5%
11. Line 436. What do they mean isolated viruses in refSeq- all viruses or a subset?
12. Line 460-476 This experiment seems self-fulfilling as using ½ data set for validation, but this was used for original discovery.

Reviewer #2 (Remarks to the Author):

The manuscript “A comprehensive metagenomic landscape of the early-life human gut virome” submitted by Zeng et al, provides a comprehensive examination of the virome in infants during their early life. The study investigates various factors linked to the infant virome, its dynamics, and the shared viral elements between infants and their mothers. Some revisions are required before publication.

1. Line 133: Please include a legend for Figure 1e, or just show one curve in each plot.
2. Line 171-175: The data showing Figure 2a do not support the statement that the diversity of infant virome increased from 0 to 36 months. Seem like a decrease was observed within 3 months. Please clarify this.
3. Line 190: The confidence interval for red color was missed in Figure 2c.
4. Line 201-205: Some viruses are believed to be contaminated by host sequences, such as mimivirus, phycodnavirus, marseillevirus, flaviviruses and poxviruses. It would be better if these families were removed from the figure and text. Please refer to this review paper (<https://doi.org/10.1038/s41579-021-00536-5>).
5. Line 332-358: Ecoli is one of the pioneer colonizers, however, the phage host prediction does not support this. Please clarify this. Maybe the host prediction is not reliable?
6. Line 460-470: Did the author consider batch effect in this analysis?

Response to the editor and reviewers

We thank the Editor for the opportunity to submit the revised manuscript “A *comprehensive metagenomic landscape of the early-life human gut virome*” (NCOMMS-23-29070-T) for publication in *Nature Communications*. We also thank all reviewers for the tremendous work, in-depth and thoughtful comments to improve our work. In the revised manuscript, all the comments and concerns from the reviewers have been addressed together with a point-by-point response as below, and changes in the revised manuscript have been highlighted in yellow.

REVIEWER COMMENTS

Reviewer #1 (Remarks to the Author):

Zeng et al. present mined publicly deposited infant stool datasets to present a catalog of infant gut viruses. Using this dataset they then attempt to better describe the infant virome over time, variation by different metadata, and shared sequences with maternal gut samples available.

Response: We sincerely appreciate the reviewer for the positive feedback on our work and also the critical drawback of the original submission, which has been revised accordingly.

Major concerns:

1. The vast majority of sequence used is derived from bulk sequenced datasets rather than VLP processed samples. Therefore, the viruses detected are most likely integrated phage. This is fine if the goal is to make a catalog, but the authors then go on to perform analyses and make conclusions about the active infant virome which may be unfounded. For example, correlations between gut virome and bacteriome don't seem surprising given the metagenome (not VLP) dataset.

Response: Thank you very much for this suggestion, and we agree with the reviewer that the bulk metagenomic sequencing, one main approach to boost the discovery of novel viral genomes^{1,2}, can simultaneously capture sequences of extracellular and intracellular viral sequences, including the integrated phages. Following the computational assembly and viral sequences prediction, distinguishing active viruses from integrated phage is challenging as viral sequences may be fragmented and therefore lack their flanking host region, which results in a failure to separate it from host sequences. Currently, this challenge can be partially addressed by CheckV, which attempts to distinguish the viral sequences to be predicted as prophages or not and removes the flanking host region. Within the ELGV catalogue, only 8.83% of viral sequences or 9.26% of vOTUs after clustering were predicted as prophages by CheckV, and the flanking host region was accordingly removed, as stated in our response below for the third minor concern. Of note, a previous study has suggested that 94.2% of prophages identified from bulk metagenomes of infants at one year of age were active³. We therefore believe that the subsequent analyses based on the ELGV catalogue are still reliable and informative.

Moreover, the current data volume of VLPs-enriched metagenomes from the gut of infants is quite limited, leading to its lower proportion (10%) compared to bulk metagenomes in our current work. We acknowledge that incorporating more VLPs-enriched metagenomes from infant gut for ELGV in future will be the continued area of investigation, so as to create an extensive, unified and standardized community resource.

To emphasize this comment and the current challenge, we have discussed this point as a limitation in the revised manuscript to merit further studies, line 583-588: *“Although the current study has advanced our understanding of the gut virome in early life both in depth and breadth, there are several limitations that need to be addressed in the future. First, the majority of gut metagenomes are from bulk sequencing, which means low-abundant viruses present in the infant gut may be underrepresented in the ELGV. Furthermore, the application of VLPs enrichment procedures may be necessary for the discovery of active viruses that do not integrate into the host genome.”*

2. RNA viral component is not addressed here which is glossed over.

Response: We thank the reviewer for this comment, and also agree that RNA virome is a key component in any ecosystem they are present in, which thus has been recently cataloged across diverse habitats on the Earth^{4,5}. However, based on previous findings from individual studies that sequenced both DNA and RNA viruses from the infant gut, RNA viruses are believed to represent a minor fraction of the gut virome in both the richness and abundance, and the majority of viruses in the infant gut are DNA phages⁶⁻⁹. The viral catalogue GVD that was built from both DNA and RNA sequencing data comprises 98% of phages and 34 OTUs (i.e., 0.10%) belonging to three viral RNA families¹⁰. Given the limited RNA sequencing data from the infant gut, we thus primarily focused on the DNA virome in current work. However, the deposited sequencing data from one study (i.e., Beller et al., 2022⁹) that was included to build the ELGV catalogue was a hybrid of DNA and cDNA (RNA) sequences, which resulted in three RNA viral families in our ELGV catalogue, including *Caliciviridae* (n = 12 vOTUs with 13 viral sequences), *Picornaviridae* (n = 6 vOTUs with 6 viral sequences), and *Astroviridae* (n = 2 vOTUs with 2 viral sequences). We have added this description in the revised manuscript, line 129-133: *“Three RNA viral families, including Caliciviridae (n = 12 with 13 viral sequences), Picornaviridae (n = 6 with 6 viral sequences), and Astroviridae (n = 2 with 2 viral sequences), were detected, which was attributed to a hybrid of DNA and cDNA (RNA) sequencing data for each sample deposited by Beller et al.³³ that was included for building the ELGV catalogue.”*

Additionally, we also discussed this point as a limitation in the revised manuscript, highlighting it for investigation, line 588-592: *“Additionally, although RNA viruses represent a minor fraction in the infant gut virome, they have been largely excluded from the ELGV catalogue due to limited metatranscriptomic sequencing data available from the infant gut. With increasing volume of viral*

sequencing data from the infant gut microbiome, an integrated and unified viral catalogue including DNA and RNA viruses from early life should be then generated.”

Minor concerns:

1. Specific parameter they used for some software are not clearly stated. For instance, on Line 591-593, they mentioned option `--keep-original-seq` for `virsorter2`, and what are the included groups and minimum score cutoff?

Response: We thank the reviewer for pointing it out and sorry to overlook these important information in the original submission. As suggested, we have added the description about the specific parameters used for each software in current work.

For instance with VirSorter2, line 622-625: *“VirSorter2 as the latest viral identifier by integrating a collection of customized automatic classifiers to improve the classification accuracy of all types of viruses, which was run with default settings (including `--include-groups dsDNAphage,ssDNA --min-score 0.5`) with additional option `'--keep-original-seq'`.”*, **and CheckV, line 627-629:** *“Thereafter, we ran CheckV v1.0.1 with `'end_to_end'` mode with default settings on all 2,864,897 viral sequences, and those ($n = 510,715$) including viral sequences and proviruses without the host contamination...”*

2. Line 576-577: They used KneadData for quality control and human contamination removal. Did the authors perform quality control on all the collected metagenomes in a uniform manner or if did they trimmed them individually based on the specific methods used to generate the metagenomes?

Response: We did run KneadData on all the metagenomes that were used in the current work for building and analysing the ELGV in a uniform manner. To make it clearer, we have rephrased the description in the revised manuscript, line 606-608: *“All collected metagenomes were quality-controlled and decontaminated of human genomic DNA (hg19 human reference genome) by KneadData v0.7.2 with default parameters.”* **and line 744-747:** *“An additional set of 301 infant (within one year old) and 510 adult gut metagenomes with >100,000 sequencing reads were obtained from five studies, respectively, and used for the classifier discovery and validation. These metagenomes were quality-controlled using KneadData with default settings.”*

3. Line 595-599: Did the authors remove the flanking host region from prophages? It would be beneficial to clarify whether the resulting catalogue includes host sequences for the identified viral sequences

Response: We acknowledge the reviewer’s suggestion. Yes, we used prophage FASTA file created by CheckV that is free from host sequences. We have modified the manuscript to clarify it, line 627-629: *“...we ran CheckV v1.0.131 with `'end_to_end'` mode with default settings on all*

2,864,897 viral sequences, and those (n = 510,715) including viral sequences and proviruses without the host contamination, which met the following criteria...”

4. Figure 1B. What does the text in the bottom `# of viral sequences` supposed to show? It seems something may be missing?

Response: We really apologize for our mistake of missing the side bar plot, which shows the number of viral sequences with various GC contents. We have added this plot in the revised manuscript.

5. Line 124 What was the other 0.2% of virus? Vertebrae viruses? Is this skewed due to the majority being non-VLP prep?

Response: Based on the voting approach, 43,271 vOTUs (64.6% of total) were taxonomically assigned at the family level based on UniProt Knowledgebase (UniProtKB), highlighting the incompleteness and knowledge gap of the current taxonomy for early-life human gut virome. Among these 43,271 taxonomic assigned vOTUs, 99.8% (n = 43,180 vOTUs) were bacteriophages, and the other vOTUs (0.2%, n = 91 vOTUs) represented eukaryotic viruses from 14 viral families, mainly including *Anelloviridae* (n = 42 vOTUs with 46 viral sequences), *Caliciviridae* (n = 12 vOTUs with 13 viral sequences), and *Adenoviridae* (n = 7 vOTUs with 23 viral sequences). All this taxonomic information has been provided in Supplementary Table 3, and the text has been rephrased, line 124-126: “Among the taxonomic assigned vOTUs in the ELGV catalogue, 99.8% (n = 43,180 vOTUs) were bacteriophages, and the other vOTUs (0.2%, n = 91 vOTUs) represented eukaryotic viruses (Supplementary Table 3).”

We speculate that the lower discovery of eukaryotic viruses was not related with the lack of VLPs enrichment and sequencing. Based on previous studies that solely sequenced VLPs-enriched metagenomes, such as Liang et al. (2020)⁷, Beller et al. (2022)⁹, and Kaelin et al. (2022)¹¹ that were included to build ELGV catalogue, they reported that the eukaryotic viruses represented a minority of relative abundance in their VLPs sequencing data.

6. Line 128. The number of microviridae seems very small.

Response: Thank you for this suggestion. We first checked our protocol for the viral identification and taxonomic assignment and ensured it correct. We then compared the proportion of *Microviridae* in ELGV catalogue to the other very recent viral catalogue CHVD¹² and MG² that however were mainly built from the adult gut metagenomes. We found a slight difference but comparable result across catalogues for vOTUs with a taxonomic assignment (0.54% of vOTUs from ELGV, 3.5% of vOTUs from CHVD, 6.0% of vOTUs from MG), which may be caused by the cut-off of the length of viral sequences (3kbp, 1.5kbp, and 1kbp, respectively) or the host age. Furthermore, based on the revised criteria for presence of vOTUs

in each metagenome, i.e., requiring $\geq 75\%$ coverage and $\geq 90\%$ identity (as response to the 8th minor comment), we found a prevalence of 13% samples having *Microviridae* family, which is also comparable with other studies involving the infant virome. For instance, Kim et al. (2022)¹³ detected *Microviridae* only in 6 out of 460 fecal samples (1.3%) from infants aging from 6 to 15 weeks of age, and Shah et al. (2023)¹⁴ did not find any members belonging to the family of *Microviridae* in the infant gut at one year old. However, Liang et al. (2020)⁷ reported an increasing prevalence of *Microviridae* of 0 or 30% of fecal samples from 20 infants at month 0 or 4 of age, respectively. Lim et al. (2015)⁶ detected the presence of *Microviridae* in all 48 fecal samples from 8 infants from six time points within the first two years of life. These results indicated the prevalence of *Microviridae* may be age- and cohort-dependent, and our result across multiple cohorts from the first three years of age falls into the reported range.

7. Line 156. What are the authors talking about regarding potential bias?

Response: Here we intended to mention the potential influences of the infant age and sequencing depth on the gut virome richness and diversity. We have modified the language by replacing “potential bias” with “influence”, line 161-163: *“Given the potential influence of the infant age and sequencing depth on the resulting gut virome composition, we further limited the infant age covered by both bulk and VLPs-enriched metagenomes...”*

8. Line 162. Why did the authors choose $>50\%$ coverage and $>95\%$ identity?

Response: We thank the reviewer for this comment, which promoted us re-think and double-check this threshold to justify the presence of the viral sequences in each metagenomes. As a result, we found that $\geq 75\%$ coverage by read mapping at $\geq 90\%$ identity were recommended for a higher sensitivity and lower false discovery rate by Roux et al. (2017)¹⁵ and used by other virome studies^{1,16}. We thus applied these criteria to re-estimate the relative abundance of viral sequences in each metagenome. The relevant text has been modified accordingly in the revised manuscript.

9. A key paper Walters et al. Cell Host Microbe. 2023 Feb 8; 31(2): 187–198.e3. is missing particularly from discussion regarding shared infant/maternal virome.

Response: Thank you for this comment. The findings from Walters et al. (2023)⁸ have been discussed and referenced in Discussion part of the revised manuscript, line 564-569: *“It has to be mentioned that the majority of gut viruses of infants we discovered belong to phages, and the shared status of eukaryotic viruses by mother-infant dyads has been largely overlooked. A recent study found distinct evolutionary patterns between phages and eukaryotic viruses in infants, where only the phage composition was found to become increasingly similar to their mothers as the infant aged. Therefore, deeper insights into the dynamics and shared status of eukaryotic viruses are needed.”* **Additionally,**

we have mentioned the continued necessity to integrate more newly sequenced VLPs-enriched data from infants to expand and create a unified and standardized community resources, line 583-588: *“Although the current study has advanced our understanding of the gut virome in early life both in depth and breadth, there are several limitations that need to be addressed in the future. First, the majority of gut metagenomes are from bulk sequencing, which means low-abundant viruses present in the infant gut may be underrepresented in the ELGV. Furthermore, the application of VLPs enrichment procedures may be necessary for the discovery of active viruses that do not integrate into the host genome.”*

10. Line 413. State extensive virome shared, but then states it is only 11.5%

Response: We have removed “extensive” from the text for a more precise description.

11. Line 436. What do they mean isolated viruses in refSeq- all viruses or a subset?

Response: We included all the viral sequences from RefSeq in this part of the analysis. The language has been modified with deleting “isolated”.

12. Line 460-476 This experiment seems self-fulfilling as using ½ data set for validation, but this was used for original discovery.

Response: Thanks for your comment. In this part, we performed two-level (i.e., discovery and independent validation cohorts) predictions using a random forest classifier. The reviewer is correct that the second half of adult metagenomes were used in both discovery and validation. However, in the discovery, this part of adult metagenomes were only used for testing the classifier, not training the classifier. Therefore, the training of the classifier in the discovery phase was completely independent. In the validation, we re-used them together with another 301 infant metagenomes (i.e., that were not used in the discovery and for building ELGV catalogue). We have modified the text in the revised manuscript to clarify this.

Reviewer #2 (Remarks to the Author):

The manuscript “A comprehensive metagenomic landscape of the early-life human gut virome” submitted by Zeng et al, provides a comprehensive examination of the virome in infants during their early life. The study investigates various factors linked to the infant virome, its dynamics, and the shared viral elements between infants and their mothers. Some revisions are required before publication.

Response: We sincerely appreciate the reviewer for the positive feedback on our work, and the manuscript has been revised accordingly.

1. Line 133: Please include a legend for Figure 1e, or just show one curve in each plot.

Response: The legend has been added as suggested.

2. Line 171-175: The data showing Figure 2a do not support the statement that the diversity of infant virome increased from 0 to 36 months. Seem like a decrease was observed within 3 months. Please clarify this.

Response: We thank the reviewer for pointing it out. We have checked the data, and agree that there was a drop in the viral richness of infants at month 3, however the overall trend of viral richness over the first three years of life is increasing, reflecting the dynamics of the viral richness. This phenomenon was also reported by a previous study, with a drop of viral richness in the first three months of life (Fig. 1E from Beller et al. 2022⁹). To address this comment, we modified the text to line 173-175: “The overall richness of the early-life gut virome increased significantly (two-sided Wilcoxon test blocked by ‘study’, $FDR < 0.0001$) as infants aged from a median number of 61 vOTUs (IQR = 40–85) at month 0 to 171 (IQR = 119–246) at month 36...”.

3. Line 190: The confidence interval for red color was missed in Figure 2c.

Response: We have checked the code to make this figure, and found that the width of the red line was set too large, while the confidence interval for the red line is quite narrow due to the very high number of comparisons between subjects, for example Extended Data Fig. 1 from Shao et al. (2019)¹⁷ and Fig. 2b from Vatanen et al. (2018)¹⁸ that also showed very narrow confidence intervals of comparisons between subjects. In the revised manuscript, we have adjusted the line width to make the confidence interval visible.

4. Line 201-205: Some viruses are believed to be contaminated by host sequences, such as mimivirus, phycodnavirus, marseillevirus, flaviviruses and poxviruses. It would be better if these families were removed from the figure and text. Please refer to this review paper (<https://doi.org/10.1038/s41579-021-00536-5>).

Response: We thank the reviewer for this comment and we have read this reference carefully. Afterwards, we first checked all the vOTUs that are taxonomically assigned to these families, and only *Phycodnaviridae* and *Mimiviridae* containing 13 viral sequences in total are present in ELGV catalogue. However, we found that other recent catalogues for the human gut virome (e.g., MGV¹⁹ and GPD¹) also recovered these families, which were kept by MGV, but removed by GPD as they were non-phage taxa. Additionally, these types of viruses are increasingly reported as members of the human gut virome^{20,21}, in particular under the unhealthy status, such as ulcerative colitis. We believe keeping the viruses from these two families will be more informative for other studies.

5. Line 332-358: Ecoli is one of the pioneer colonizers, however, the phage host prediction does not support this. Please clarify this. Maybe the host prediction is not reliable?

Response: Thank the reviewer for this comment. We have checked the predicted host of all vOTUs, and did find 97 vOTUs assigned to *E. coli*. We added this in the revised manuscript, line 325-326: “Of note, 97 vOTUs were predicted to be hosted by *Escherichia coli*, the species represented by the largest number of genomes in the ELGG catalogue.”

6. Line 460-470: Did the author consider batch effect in this analysis?

Response: Thank you for this comment. In order to address the batch effect in this part of analysis, we randomly selected half of the samples from each study, and then combined them into the dataset for training the classifier, and the other half for testing the classifier. The text has been modified in the revised manuscript, line 473-474: “...half of 510 adult metagenomes and half of infant metagenomes that were used to build the ELGV catalogue were randomly selected from each study and then combined for training...”

References

- 1 Camarillo-Guerrero, L. F., Almeida, A., Rangel-Pineros, G., Finn, R. D. & Lawley, T. D. Massive expansion of human gut bacteriophage diversity. *Cell* **184**, 1098-1109.e9 (2021).
- 2 Nayfach, S. et al. Metagenomic compendium of 189,680 DNA viruses from the human gut microbiome. *Nat. Microbiol.* **6**, 960-970 (2021).
- 3 Tamsin, A. R. et al. Prophages in the infant gut are largely induced, and may be functionally relevant to their hosts. *bioRxiv* (2021). doi: <https://doi.org/10.1101/2021.06.25.449885>
- 4 Neri, U. et al. Expansion of the global RNA virome reveals diverse clades of bacteriophages. *Cell* **185**, 4023-4037.e18 (2022).
- 5 Zayed, A. A. et al. Cryptic and abundant marine viruses at the evolutionary origins of Earth's RNA virome. *Science* **376**, 156-162 (2022).
- 6 Lim, E. S. et al. Early life dynamics of the human gut virome and bacterial microbiome in infants. *Nat. Med.* **21**, 1228-1234 (2015).
- 7 Liang, G. et al. The stepwise assembly of the neonatal virome is modulated by breastfeeding. *Nature* **581**, 470-474 (2020).
- 8 Walters, W. A. et al. Longitudinal comparison of the developing gut virome in infants and their mothers. *Cell Host Microbe* **31**, 187-198.e3 (2023).
- 9 Beller, L. et al. The virota and its transkingdom interactions in the healthy infant gut. *Proc. Natl. Acad. Sci. U. S. A.* **119**, e2114619119 (2022).
- 10 Gregory, A. C. et al. The gut virome database reveals age-dependent patterns of virome diversity in the human gut. *Cell Host Microbe* **28**, 724-740.e8 (2020).

- 11 Kaelin, E. A. *et al.* Longitudinal gut virome analysis identifies specific viral signatures that precede necrotizing enterocolitis onset in preterm infants. *Nat. Microbiol.* **7**, 653-662 (2022).
- 12 Tisza, M. J. & Buck, C. B. A catalog of tens of thousands of viruses from human metagenomes reveals hidden associations with chronic diseases. *Proc. Natl. Acad. Sci. U. S. A.* **118**, e2023202118 (2021).
- 13 Kim, A. H. *et al.* Enteric virome negatively affects seroconversion following oral rotavirus vaccination in a longitudinally sampled cohort of Ghanaian infants. *Cell Host Microbe* **30**, 110-123.e5 (2022).
- 14 Shah, S. A. *et al.* Expanding known viral diversity in the healthy infant gut. *Nat Microbiol* **8**, 986-998 (2023).
- 15 Roux, S., Emerson, J. B., Eloë-Fadros, E. A. & Sullivan, M. B. Benchmarking viromics: an in silico evaluation of metagenome-enabled estimates of viral community composition and diversity. *PeerJ* **5**, e3817 (2017).
- 16 Yang, Y. *et al.* A novel alteromonas phage lineage with a broad host range and small burst size. *Microbiol. Spectr.* **10**, e0149922 (2022).
- 17 Shao, Y. *et al.* Stunted microbiota and opportunistic pathogen colonization in caesarean-section birth. *Nature* **574**, 117-121 (2019).
- 18 Vatanen, T. *et al.* The human gut microbiome in early-onset type 1 diabetes from the TEDDY study. *Nature* **562**, 589-594 (2018).
- 19 Nayfach, S. *et al.* CheckV assesses the quality and completeness of metagenome-assembled viral genomes. *Nat. Biotechnol.* **39**, 578-585 (2021).
- 20 Moreno-Gallego, J. L. *et al.* Virome diversity correlates with intestinal microbiome diversity in adult monozygotic twins. *Cell Host Microbe* **25**, 261-272.e5 (2019).
- 21 Zuo, T. *et al.* Gut mucosal virome alterations in ulcerative colitis. *Gut* **68**, 1169-1179 (2019).

Reviewers' comments:

Reviewer #1 (Remarks to the Author):

Major concerns:

1. Zeng et al. state they aim to provide a catalog of early life gut virome, but neglect to include large virome (VLP) based studies in their database and subsequent analysis. For example, Shah et al., Nature Micro 2023 includes 647 1 year old viromes and Walters et al, CHM 2023 includes >450 stools from a longitudinal cohort of infants and their mothers, as well as Lim et al. 2015, Zhao et al. 2017, Magsood et al. 2019, Kim et al. 2022. What is the rationale for not being as comprehensive as possible if the aim is to make a reference catalog?
2. If the goal is to serve as an early life catalog, they should make clear which of the studies they chose to use were included in the other databases such as GVD, GPD, MGVD, and CHVD. It needs to be spelled out clearly for the reader what is overlapped.
3. Regarding the RNA viral- Walters et al. report a significant RNA phage and RNA vertebrae virus component. To not include it in this catalogue will lead to future studies also underreporting the RNA component. They should make clear which studies they chose to include represent DNA or DNA and RNA component. The lack of RNA component is a major weakness and pointing back to other catalogues which also under-represented RNA data is not a reason to propagate this under-representation.
4. Bulk vs VLP data. This is a major issue. The preprint they mention (which was posted in 2021 and still not published) is very much in contrast to the findings of Gregory et al. CHM 2020 which demonstrates about 10% overlap between VLP and bulk data suggesting that the two methods do not reflect the same communities. Therefore, all of the analyses and conclusions the authors draw about the active infant virome could be unfounded here. Furthermore, it is a missed opportunity in the current version that they do not explore the comparison of VLP to bulk data more.

Minor concerns:

1. Details regarding if RCA, random amplification, or WGA were used should be included in their summary table of the samples used as each amplification method brings its own biases and this needs to be clear to the reader.
2. Line 69 does not reference Walters et al. which as mentioned in the first review is the largest infant-mom gut virome paper to date that we are aware of.
3. Authors rebuttal to inclusion of phycodna and mimi is not sufficient. Please examine the vOTUs that you have for these families and ensure they are truly hitting virus and are not a false positive

which is often the case for these viral families. Merely saying they are of low abundance and are represented in other catalogues is not a sufficient exploration into these suspect sequences.

Reviewer #2 (Remarks to the Author):

All my comments have been addressed.

Response to the editor and reviewers

We thank both reviewers for the tremendous work, in-depth and thoughtful comments to improve our work. In the revised manuscript, all the comments and concerns from the reviewers have been addressed together with a point-by-point response as below, and changes in the revised manuscript are highlighted in yellow.

REVIEWER COMMENTS

Reviewer #1 (Remarks to the Author):

Major concerns:

1. Zeng et al. state they aim to provide a catalog of early life gut virome, but neglect to include large virome (VLP) based studies in their database and subsequent analysis. For example, Shah et al., *Nature Micro* 2023 includes 647 1 year old viromes and Walters et al, *CHM* 2023 includes >450 stools from a longitudinal cohort of infants and their mothers, as well as Lim et al. 2015, Zhao et al. 2017, Magsood et al. 2019, Kim et al. 2022. What is the rationale for not being as comprehensive as possible if the aim is to make a reference catalog?

Response: We very much appreciate the reviewer's suggestion to include the cited studies, which will comprehensively increase the coverage of our catalogue. As suggested, we searched for all studies of the infant gut virome in PubMed with terms "(infant) AND ((gut) OR (enteric) OR (intestine)) AND (virome)" in October 2023 together with manual curation. After manually checking the availability of the sequencing data and whether they correctly matched with the metadata reported in publications, we finally included 9 studies with 1,865 VLPs-enriched metagenomes, which is much higher than the sample size of 647 VLPs in the last version of manuscript. All studies mentioned by the reviewer were included in the revised manuscript, except for Walters et al, 2023, Lim et al. 2015, and Kim et al. 2022 due to the following reasons. For Walters et al, 2023 and Lim et al. 2015, only single-end reads have been deposited in NCBI, which is inconsistent with their reported sequencing method (i.e., paired-end reads) in their respective publications. We thus contacted the corresponding authors in October 2023: the author of Walters et al, 2023 confirmed the incompleteness of the deposited sequencing data, which has not yet been resolved; and no response was received from Lim et al. 2015. Lastly, the work from Kim et al. 2022 was also not included as no sequencing data is currently available in ENA or NCBI under project PRJEB39845. Therefore, we unfortunately have to exclude the three studies in the revised manuscript. Nevertheless, a total of 825 VLPs-enriched metagenomes for RNA or both DNA/RNA viruses have been included in the revised manuscript to catalogue the RNA viral sequences, which addresses the concern #3 raised by the reviewer as below.

Meanwhile, 143 bulk metagenomes from one of included studies (Liang et al., 2020, *Nature*) were newly added as these samples were processed by using both VLPs and bulk metagenomic

sequencing for the analyses of “Host prediction and close interactions between the gut virome and bacteriome early in life”. Together with all bulk metagenomes (n = 6,122) that were used to build the catalogue of the early-life human gut genomes (ELGG catalogue, Zeng et al., 2022, *Nature Communications*), we finally included 8,130 (1,865 for VLPs and 6,265 for bulk) in this revised manuscript for mining the DNA and RNA viral sequences from the human early in life. A number of 26 VLP-enriched metagenomes from mothers (Maqsood et al., 2019, *Microbiome*) whose infants were included in the 1,865 VLPs-enriched metagenomes were also added for the analyses of “A set of viruses shared by paired mother-infant dyads”. The updated ELGV catalogue and representatives reported in this revised manuscript are available in the link: <https://doi.org/10.6084/m9.figshare.21901557.v2>.

The rationale for study inclusion has also been clarified in the revised manuscript, line 701-709: “PubMed with terms “(infant) AND ((gut) OR (enteric) OR (intestine)) AND (virome)” were combined to search studies that included fecal virome sequencing data from infants (up to October 2023). Datasets were subsequently manual curated to remove studies that did not correctly match the relevant metadata or did not have any sequencing data available. After this selection process, nine studies including 1,865 VLPs-enriched metagenomes were retrieved. In addition, 143 bulk metagenomes were added from one of the nine studies (Liang et al.¹⁵) as these samples were processed by using both VLPs and bulk metagenomic sequencing. We further used all bulk metagenomes (n = 6,122) that were used to build the catalogue of early-life human gut genomes (i.e., the ELGG catalogue, Zeng et al.⁴¹) for mining the viral sequences (Supplementary Table 1).”

2. If the goal is to serve as an early life catalog, they should make clear which of the studies they chose to use were included in the other databases such as GVD, GPD, MGV, and CHVD. It needs to be spelled out clearly for the reader what is overlapped.

Response: Thank you very much for this suggestion. We have compared studies included for ELGV catalogue to the other four viral catalogues that were mainly reconstructed with adult fecal metagenomes. The result has been added into Supplementary Table 10 and the main text, line 500-502: “Compared to the metagenomes that were used to generate other existing viral databases (i.e., CHVD, GPD, GVD, and MGV), 5,068 metagenomes (1,089 for VLPs and 3,979 for bulk) distributed across 20 studies were unique to the ELGV catalogue (Supplementary Table 10).”

3. Regarding the RNA viral- Walters et al. report a significant RNA phage and RNA vertebrae virus component. To not include it in this catalogue will lead to future studies also underreporting the RNA component. They should make clear which studies they chose to include represent DNA or DNA and RNA component. The lack of RNA component is a major weakness and pointing back to other catalogues which also under-represented RNA data is not a reason to propagate this under-representation.

Response: We sincerely appreciate the reviewer for this suggestion, and apologize for our unsatisfied response in the first round of review. In this revised manuscript, we have re-searched for all the studies of gut virome in infants, and included 1,865 VLPs-enriched metagenomes from 9 studies. Unfortunately, as explained above, we were not able to include the data from Walters et al, 2023. However, among the 1,865 VLPs-enriched metagenomes, 825 samples were sequenced for RNA or both DNA/RNA viruses, and this information has been added into the curated metadata (Supplementary Table 1) as suggested. Among the 46 viral families annotated across the whole ELGV catalogue, three RNA viral families, including *Caliciviridae* (n = 14 with 16 viral sequences), *Picornaviridae* (n = 10 with 10 viral sequences), and *Astroviridae* (n = 3 with 3 viral sequences), were obtained, which were only detected in VLPs-enriched metagenomes for RNA viruses. Moreover, the three viral families of RNA were consistent with the findings from Walters et al, 2023 with *Picornaviridae* as the dominant and prevalent individual virus species of RNA¹.

4. Bulk vs VLP data. This is a major issue. The preprint they mention (which was posted in 2021 and still not published) is very much in contrast to the findings of Gregory et al. CHM 2020 which demonstrates about 10% overlap between VLP and bulk data suggesting that the two methods do not reflect the same communities. Therefore, all of the analyses and conclusions the authors draw about the active infant virome could be unfounded here. Furthermore, it is a missed opportunity in the current version that they do not explore the comparison of VLP to bulk data more.

Response: We thank the reviewer and agree with this comment. In the original manuscript, we did not compare the VLPs-enriched and bulk metagenomes, due to the limited sample size of VLPs. In this revised manuscript with larger number of VLPs-enriched metagenomes included, we simultaneously conducted all the analyses that have been applied to bulk metagenomes on VLPs-enriched metagenomes, which comprehensively revealed the properties of the infant gut virome under different circumstances of sequencing approaches.

First, we indeed observed some comparable trends of the infant gut virome profiled by VLPs and bulk, such as the increased alpha diversity and decreased inter-individual variability as infant aged, although the richness of vOTU representatives from bulk was higher but the viral variability among samples was lower than that of VLPs. Both approaches also revealed interactions between infant gut virome and bacteriome, and the presence of mother-infant shared viruses, but with varying degrees.

Regarding the viral composition profiled by both approaches, there was a proportion of ~40% vOTU representatives that overlapped by VLPs and bulk based on either the same fecal samples from Liang et al. 2020² or the large combined population. This proportion is higher than 8.5% for the same adult fecal samples or 10% GVD viral populations reported by Gregory et al. 2020³. We propose that this inconsistency is most likely attributed to the host age, as a

relatively simple gut virome is present in early life and gradually becomes diverse in later life⁴. At the family level, 9 viral families that were absent in bulk were detected in VLPs, including all three RNA viral families *Astroviridae*, *Caliciviridae*, and *Picornaviridae* as expected. There were also four viral families that were detected by bulk but missed in VLPs, including *Anaerodiviridae*, *Mesyanzhinovviridae*, *Vilmaviridae*, and *Zierdtviridae* (Supplementary Fig. 4b). Moreover, the viral family in VLPs-enriched metagenomes was mainly dominated by *Microviridae* and *Siphoviridae*, which was highly consistent with the findings from Walters et al. 2023¹; whereas the abundance of *Microviridae* was much lower in bulk metagenomes. The prevalence of *Microviridae* was also much higher in VLPs-enriched metagenomes (65.6% of 1,865 samples) than bulk (9.48% 6,265 samples). We also found the infant gut virome from VLPs showed a weaker association to clinical factors (e.g., delivery mode and gestational age) compared to that of bulk, and an increased number of predictive features were required to distinguish the gut virome from infants and adults (AUC = 0.84 with 64 vOUTs from VLPs and AUC > 0.97 with as few as 8 vOTUs from bulk). We propose that this may be attributed to the larger variability of gut virome profiled by VLPs than that of bulk.

Overall, in the revised manuscript with the inclusion of more VLPs-enriched metagenomes, we have conducted a comprehensive comparison between VLPs and bulk metagenomic sequencing to characterize the gut virome in early life. The manuscript has been accordingly revised, and some key changes are listed as below.

Revised text in Results:

Line 164-168: “A previous study has indicated differences in the composition of gut virome from VLPs-enriched or bulk metagenomes, as the former captures infecting viruses or integrated prophages while the latter targets free or active viruses¹⁶. We therefore analyzed VLPs-enriched or bulk metagenomes separately thereafter to comprehensively reveal properties of the early-life gut virome by different sequencing approaches.”

Line 178-194: “A total of 28,531 and 64,934 vOTUs were respectively detected in 1,682 VLPs-enriched and 6,205 bulk metagenomes, with a median relative abundance of 218 and 298 reads per kilobase per million mapped reads (RPKM) (IQR = 99–564 RPKM for VLPs and IQR = 111–854 RPKM for bulk), which were used for subsequent analyses. We found that 11,458 vOTUs were recovered by both approaches, indicating that 40.2% of vOTUs from VLPs-enriched metagenomes could be captured with bulk metagenomic sequencing (Supplementary Fig. 2c). Furthermore, based on fecal samples that were processed by both VLPs-enriched and bulk metagenomic sequencing from Liang et al.¹⁵, 2,234 vOTUs were captured by both approaches, accounting for 41.7% of VLPs and 34.7% of bulk (Supplementary Fig. 2c). The richness of vOTUs from bulk metagenomes was found to be higher (two-sided Wilcoxon test blocked by ‘study’, $P < 0.0001$) than that of VLPs-enriched metagenomes (median = 79; IQR = 50–130 for bulk; median = 32; IQR = 8–81 for VLPs). Both sequencing approaches showed great viral variability among samples as only small parts of vOTUs

were populated with a prevalence $\geq 5\%$ ($n = 66$ for VLPs and 284 for bulk) and 1% ($n = 990$ for VLPs and 1,561 for bulk) (Supplementary Fig. 2d,e). Moreover, the gut virome profiled by VLPs exhibited higher (two-sided Wilcoxon test, $P < 0.0001$) variability than that of bulk based on these vOTUs at a prevalence $\geq 5\%$ or 1%, and this was also confirmed by analyzing the fecal samples that were processed by both VLPs-enriched and bulk metagenomes¹⁵.”

Line 218-232: “To characterize the composition of the early-life human gut viral community, we summed the relative abundance of individual vOTU representative sharing the same family rank for each metagenome to reflect the viral temporal changes at the family level. A total of 42 viral families from 1,682 VLPs-enriched metagenomes with ≥ 1 vOTU detected accounted for 87.9% of total abundance (median, IQR = 61.6–98.8%); while 37 families from bulk metagenomes accounted for a lower relative abundance (median = 41.4%, IQR = 31.3–53.6%), indicating more unclassified viruses were captured by bulk metagenomes (Supplementary Fig. 4a). Notably, nine viral families that were absent in bulk metagenomes were detected in VLPs-enriched metagenomes, including all three RNA viral families Astroviridae, Caliciviridae, and Picornaviridae. There were also four viral families that were detected by bulk metagenomes but missed in VLPs-enriched metagenomes, including Anaerodiviridae, Mesyanzhinoviridae, Vilmaviridae, and Zierdtviridae (Supplementary Fig. 4b). In VLPs-enriched metagenomes, the most abundant families in the first two years of life included Microviridae, Siphoviridae, and Myoviridae, however, Microviridae had a lower (two-sided Wilcoxon test, $P < 0.05$) relative abundance in bulk metagenomes that were dominated by Siphoviridae, Myoviridae, and Peduoviridae (Supplementary Table 5).”

Line 232-244: “To further explore the dynamics of individual viral families early in life, we partitioned the fecal samples into discrete time points if infant age at sampling available and assessed their variation using linear mixed modelling. It was observed that the succession of an individual viral family varied over time, with 7 and 21 families changing significantly from VLPs-enriched and bulk metagenomes, respectively (linear mixed modelling with ‘study’ as random factor, $P < 0.05$; Supplementary Table 5). Out of 42 viral families with a prevalence $> 1\%$, 25 families accounted for $> 99\%$ of viral abundance in 1,608 VLPs-enriched metagenomes. Of these, five families with significant changes and only Microviridae increased in the abundance as infants aged (Fig. 2d). When examining these 25 viral families in bulk metagenomes, 21 families were detected and accounted for $> 99\%$ of viral abundance in 5,990 bulk metagenomes. Of these, 14 families had a statistically significant difference ($P < 0.05$) in their abundance as infants aged, with half of the families increased, such as Adenoviridae, Duneviridae, and Forsetiviridae; and the other half decreased, such as Siphoviridae, Myoviridae, Microviridae, and Peduoviridae.”

Line 313-321: “We did not observe any families significantly ($q < 0.25$) associated with delivery mode, gestational age, and feeding pattern at sampling based on VLPs-enriched metagenomes, while some significant families were identified from bulk metagenomes (Fig. 3c-e). More specifically, bulk metagenomes from infants born by C-section (taking vaginal delivery as reference in MaAsLin2

model) were enriched ($q < 0.25$) with the viral families Herelleviridae and Podoviridae although they only accounted for an average of 1.39% in relative abundance of the early-life human gut virome. In contrast, infants born vaginally harbored higher ($q < 0.25$) abundance of Microviridae, Autographiviridae, Forsetiviridae, and Siphoviridae, and these families comprised an average of 39.1% in relative abundance (Fig. 3c).”

Line 364-370: “Given the impact of the gut virome diversity on the gut bacteriome and vice-versa in adults^{11,43}, we examined the temporal correlation between the composition of early-life human gut virome and bacteriome, which thus far is not clearly understood. To address this, we focused on the bacteriome of 141 fecal samples that were simultaneously processed by using both VLPs-enriched and bulk metagenomic sequencing from Liang et al.¹⁵ and 6,066 fecal samples that were only subjected to bulk metagenomic sequencing to compare with the corresponding virome in each sample according to different sequencing approaches.”

Line 416-422: “To fully decipher the shared and unique properties of mother-infant virome particularly early in life, we analyzed all fecal samples from mothers whose infants were included in the reconstruction of the ELGV catalogue. As a result, 373 paired mother-infant dyads including 460 maternal fecal bulk metagenomes (covering pregnancy, delivery, and postpartum) and 1,000 infant fecal bulk metagenomes (ranging from birth to the first two years of life) were analyzed (Supplementary Table 8). Additionally, we also collected the VLPs-enriched metagenomes from 26 paired mother-infant dyads from Maqsood et al.²⁷, including 26 maternal and 48 infant samples (Supplementary Table 8).”

Line 430-444: “Based on VLPs-enriched metagenomes, we clustered 656 qualified mother-infant viral sequences into 511 vOTUs at 95% average nucleotide identity (ANI) over 85% alignment fraction (AF)³³. Among them, 199 and 307 vOTUs were exclusively represented by viral sequences from mothers and infants, respectively, and only five vOTUs were shared by mothers and their paired or unpaired infants (Fig. 5a). After excluding vOTUs exclusively containing viral sequences from the unpaired mother-infant dyads, four vOTUs were shared by two paired mother-infant dyads (hereafter referred to as shared-vOTUs; Fig. 5a), belonging to Siphoviridae ($n = 3$) and Podoviridae ($n = 1$). Apart from the assembly-based approach, we further mapped the quality-controlled from mothers and infants to ELGV representatives to calculate the relative abundance of each vOTU representative. We found that maternal gut viromes differed from infant gut viromes based on Bray–Curtis distances (PERMANOVA, 1,000 permutations, $P < 0.001$) but the average of richness between mothers and infants was comparable (two-sided Wilcoxon test, $P = 0.84$), which was consistent with reports from the original publication²⁷ (Fig. 5b). Only two paired mother-infant dyads were found to share vOTUs from the reads-based mapping, and one pair of them (C047) was also observed from the assembly-based approach.”

Line 516-528: “Whether viruses that are specifically present in early life remains open. Considering the high inter-individual variability in the early-life human gut virome, we primarily

focused on the vOTUs if their prevalence exceeded 2% across 1,682 VLPs-enriched metagenomes with a relative abundance > 0.01 (1%) in at least one metagenome, while the threshold values were ten-fold increased for bulk metagenomes up to > 20% and > 0.10 given there was a lower inter-individual variability and larger sample size (n = 6,205). This resulted in 407 vOTUs from VLPs-enriched metagenomes, which accounted for an average relative abundance of 33.7% (median = 21.2%; IQR = 1.77–62.3%) with a prevalence ranging from 2.02% to 20.1%, and 335 vOTUs exclusively belonged to the ELGV catalogue. To check their specificity for the early-life human gut virome, we quantified their relative abundance in 521 adult VLPs-enriched metagenomes (Supplementary Table 11), and found that 111 of 139 significantly differential vOTUs (MaAsLin2 with age (infants vs. adults) as fixed effect, $q < 0.25$) were lower in adults than that of infants (an average of relative abundance of 6.58% vs. 17.4%; Fig. 6b).”

Revised text in Discussion:

Line 662-681: *“Currently, two main approaches exist for studying the viral profile within a microbiome: VLPs-enriched metagenomic sequencing and bulk metagenomic sequencing. Both approaches have pros and cons. For instance, VLPs-enriched metagenomes may skew viral profiles and abundances due to incomplete removal of cellular organisms, and exclude large viruses by size filtration and whole-genome amplification. On the other hand, the use of bulk metagenomes may miss low abundance viruses, especially if samples are not sequenced at sufficient depth^{18,57}. By conducting similar analyses from VLPs-enriched and bulk metagenomes separately, we observed comparable results in profiling of the human gut virome early in life, such as the increased alpha diversity and decreased inter-individual variability. There was a proportion of ~40% viruses that overlapped by VLPs-enriched and bulk metagenomes sequenced from either the same fecal samples or the combined metagenomes, which is higher than the proportion (~10%) previously estimated in adults¹⁶. We propose that this inconsistency may be attributed to the host age, as a relatively simple gut virome was found early in life and gradually becomes diverse later in life⁴. Additionally, the RNA viral families were as expected only detected in VLPs-enriched metagenomes. Viral families from VLPs-enriched metagenomes were mainly dominated by Microviridae and Siphoviridae, which is consistent with the findings from Walters et al.²⁸; whereas the abundance of Microviridae was much lower in bulk metagenomes. Moreover, VLPs-enriched metagenomes revealed larger viral variability between samples, which may partly explain the lower association of the gut virome with clinical factors (e.g., delivery mode and gestational age) and an increased number of predictive features that were necessary to distinguish the gut virome from infants and adults.”*

Minor concerns:

1. Details regarding if RCA, random amplification, or WGA were used should be included in their summary table of the samples used as each amplification method brings its own biases and this needs to be clear to the reader.

Response: The amplification approach used in each of nine studies has been added into Supplementary Table 1, including MDA for four studies, SIA for three studies, and MDA and SIA for two studies.

2. Line 69 does not reference Walters et al. which as mentioned in the first review is the largest infant-mom gut virome paper to date that we are aware of.

Response: We agree with the reviewer, and the study Walters et al, 2023 has been referenced here as suggested.

3. Authors rebuttal to inclusion of phycodna and mimi is not sufficient. Please examine the vOTUs that you have for these families and ensure they are truly hitting virus and are not a false positive which is often the case for these viral families. Merely saying they are of low abundance and are represented in other catalogues is not a sufficient exploration into these suspect sequences.

Response: Thank you very much for this suggestion. In the revised manuscript, we have now excluded the families *Phycodnaviridae*, *Mimiviridae* and *Marseilleviridae* from the ELGV catalogue, as we agree that they may represent contaminants or misannotations, as suggested by the reviewer and others^{3,5}.

Line 106-112: “Notably, we found that 11 vOTUs were assigned to the viral families of *Phycodnaviridae* ($n = 5$ with 8 viral sequences), *Mimiviridae* ($n = 4$ with 8 viral sequences), and *Marseilleviridae* ($n = 2$ with 4 viral sequences), which likely represent contaminants or misclassifications, as previously suggested^{16,34}. Therefore, we manually removed these vOTUs and their viral sequences, resulting in 82,141 vOTUs containing 160,478 viral sequences, henceforth referred to as the ELGV catalogue (Fig. 1b; Supplementary Table 2 for viral sequences, Supplementary Table 3 for vOTUs).”

Reviewer #2 (Remarks to the Author):

All my comments have been addressed.

Response: We thanks the reviewer for the positive feedback.

References

- 1 Walters, W. A. *et al.* Longitudinal comparison of the developing gut virome in infants and their mothers. *Cell Host Microbe* **31**, 187-198.e3 (2023).
- 2 Liang, G. *et al.* The stepwise assembly of the neonatal virome is modulated by breastfeeding. *Nature* **581**, 470-474 (2020).

- 3 Gregory, A. C. et al. The gut virome database reveals age-dependent patterns of virome diversity in the human gut. *Cell Host Microbe* **28**, 724-740.e8 (2020).
- 4 Liang, G., Gao, H. & Bushman, F. D. The pediatric virome in health and disease. *Cell Host Microbe* **30**, 639-649 (2022).
- 5 Liang, G. & Bushman, F. D. The human virome: assembly, composition and host interactions. *Nat. Rev. Microbiol.* **19**, 514-527 (2021).

REVIEWERS' COMMENTS

Reviewer #1 (Remarks to the Author):

The authors made significant changes to address concerns raised on review.

In general the text is dense and at time hard to follow.

Minor comments:

1. It is unclear what was done with the vOTUs that clustered to RefSeq. What was done if a vOTU clustered to RefSeq and by the other methods used? What was used?
2. Few RNA viruses. Surprised no tombusviridae.
3. Line 442. Only two paired mother-infant dyads were found to share vOTUs from the reads-based mapping, and one pair of them (C047) was also observed from the assembly based approach. This is different than the original study. Why do the authors think this might be?
4. Fig 2d what do the different colors mean (red vs green)?

Response to the editor and reviewers

We thank the Editor for the opportunity to submit the revised manuscript for publication in *Nature Communications*. We also thank the reviewer for the tremendous work, in-depth and thoughtful comments to improve our work. In the revised manuscript, all the comments and concerns from the reviewer have been addressed together with a point-by-point response as below, and changes in the revised manuscript have been highlighted in yellow.

REVIEWERS' COMMENTS

Reviewer #1 (Remarks to the Author):

The authors made significant changes to address concerns raised on review.

Response: We sincerely appreciate the reviewer for the positive feedback on the revised manuscript.

In general the text is dense and at time hard to follow.

Response: Thank you for your advice. Accordingly, we have modified the manuscript by shortening, rephrasing or deleting some of the text to make the manuscript clearer and easier to follow. For example, we deleted some results that were repeated or partially overlapped, such as the content in line 91-93 in current version has been shortened by deleting the description for each viral identifier as they have been given in the Methods; line 148-156 (in the last version of manuscript) as mentioned by the reviewer from #1 minor comment, and line 232-234 (in the last version of manuscript) for the description how to partition the fecal samples into discrete time points. In addition, some of the interpretation of the data that was repeated both in Results and Discussion have been removed from the Results section to make it more concise, such as line 214-215 and line 282-284 (in the last version of manuscript).

Minor comments:

1. It is unclear what was done with the vOTUs that clustered to RefSeq. What was done if a vOTU clustered to RefSeq and by the other methods used? What was used?

Response: In the manuscript, we always prioritized using the vOTU sequences from the ELGV catalogue for all subsequent analyses as this represented a closer match to the sequences in the metagenomes. With this clustering analysis, we basically intended to examine the novelty of the vOTU sequences in comparison to viral sequences deposited in NCBI RefSeq. During revising the manuscript, we realized that this purpose has been achieved in the other part of analyses from line 470-486 (in current version) for cross-catalogue comparisons (i.e., CHVD, GPD, GVD, MGV, and RefSeq). Therefore, we deleted this part of content (line 148-156 in the last version of manuscript) to make the manuscript concise. Thank you for pointing it out.

2. Few RNA viruses. Surprised no tombusviridae.

Response: Based on the reviewer's comment, we further investigated potential reasons for the lack of *Tombusviridae* viruses in our catalogue. On the one hand, we checked the protein database (UniProt Knowledgebase including TrEMBL and Swiss-Prot; Release 2022_03) that was used for the taxonomic annotation, and confirmed that the protein sequences from this viral family represent a very limited proportion of the database (0.1% or 5363 out of 5,230,664 sequences in total). On the other hand, we compared our pipeline to that of Walters et al. (PMID: 36758519) that detected this family in infants, and found that they employed a distinct, read-based approach without contig assembly. We thus suppose that both our filtering criteria for viral contigs and the limited number of representatives in the UniProt database led to a lack of detection of *Tombusviridae* sequences.

3. Line 442. Only two paired mother-infant dyads were found to share vOTUs from the reads-based mapping, and one pair of them (C047) was also observed from the assembly based approach. This is different than the original study. Why do the authors think this might be?

Response: Thank you for the question, and we indeed observed fewer number of mother-infant dyads with shared viruses compared to Maqsood et al. (PMID: 31823811). We believe these differences are mainly due to differences in the pipeline used for viral detection. Unlike our pipeline, which integrated multiple filtering criteria based on the quality metrics from CheckV and viral contigs longer than 3000 bp, Maqsood et al. used a lower length threshold (contigs longer than 500 bp) and did not apply any additional tools to check the quality of the viral contigs. Furthermore, in the mapping process we also excluded reads that mapped with a percent identity < 90%, and only considered a viral contig to be present if it had a minimum coverage $\geq 75\%$. Therefore, we expect that our stricter selection for viral contigs and mapping resulted in a decreased number of viral contigs present in mother or infant samples, thereby leading to a reduced number of shared viral contigs.

4. Fig 2d what do the different colors mean (red vs green)?

Response: The colors (i.e., red, light green, and dark green) correspond to the viral families with different mean relative abundances. We have added this explanation in the figure legend, line 1072-1076: "*For better visualization of the changes of each viral family, viral families are stratified into three groups based on the mean relative abundance of VLPs-enriched metagenomes at each time point (i.e., maximal mean relative abundance $\leq 1\%$ (left, $n = 12$, red), maximal mean relative abundance $> 1\%$ and $< 40\%$ (middle, $n = 11$, light green), maximal mean relative abundance $\geq 40\%$ (right, $n = 2$, dark green)).*"